# Physico-Chemical Characterization, Phenolic Compound Extraction and Biological Activity of Grapevine (*Vitis vinifera* L.) Canes

Răzvan Vasile Filimon [1,†], Claudiu Ioan Bunea [2,†], Florin Dumitru Bora [3,4], Roxana Mihaela Filimon [1,*], Simona Isabela Dunca [5], Sándor Rózsa [6], Liliana Ciurlă [7] and Antoanela Patraș [7]

1 Research Development Station for Viticulture and Winemaking Iasi, 48 Mihail Sadoveanu Alley, 700490 Iasi, Romania; razvan_f80@yahoo.com
2 Department of Viticulture and Oenology, Faculty of Horticulture and Business in Rural Development, University of Agricultural Sciences and Veterinary Medicine Cluj-Napoca, 3-5 Mănăștur Street, 400372 Cluj-Napoca, Romania; claudiu.bunea@usamvcluj.ro
3 Department of Viticulture and Oenology, Advanced Horticultural Research Institute of Transylvania, Faculty of Horticulture and Business in Rural Development, University of Agricultural Sciences and Veterinary Medicine Cluj-Napoca, 3-5 Mănăștur Street, 400372 Cluj-Napoca, Romania; boraflorindumitru@gmail.com
4 Laboratory of Chromatography, Advanced Horticultural Research Institute of Transylvania, Faculty of Horticulture and Business in Rural Development, University of Agricultural Sciences and Veterinary Medicine Cluj-Napoca, 3-5 Mănăștur Street, 400372 Cluj-Napoca, Romania
5 Faculty of Biology, "Al. I. Cuza" University of Iasi, 11 Carol I Boulevard, 700506 Iasi, Romania; sdunca@uaic.ro
6 Horticultural Products Technology-Research Department, Faculty of Horticulture and Business in Rural Development, University of Agricultural Sciences and Veterinary Medicine Cluj-Napoca, 3-5 Mănăștur Street, 400372 Cluj-Napoca, Romania; rozsa.sandor@usamvcluj.ro
7 Faculty of Horticulture, "Ion Ionescu de la Brad" Iasi University of Life Sciences (IULS), 3 Mihail Sadoveanu Alley, 700490 Iasi, Romania; liliana.lucescu@yahoo.com (L.C.); apatras@uaiasi.ro (A.P.)
* Correspondence: roxanacotovanu@yahoo.com
† These authors contributed equally to this work.

**Abstract:** Annual grapevine pruning produces large amounts of unused waste as woody canes. The current study is aimed at the sustainable valorization of viticultural waste by establishing phenolic compound extraction conditions, composition, and biological potential of crude and purified cane extracts of three *Vitis vinifera* L. cultivars growing in temperate climate conditions. Grapevine canes proved to be rich in carbohydrates and minerals; chlorophyll and carotenoids were also quantified. The highest yield of phenolic compounds was obtained when dry canes (<0.5 mm) were subjected to liquid–solid extraction (1:20 *w/v*) with 70% (*v/v*) ethanol, for 4 h at 35 °C, after a preliminary ultrasound treatment (6 min., 42 KHz); Pinot Gris canes showing the highest concentration of flavonoids and non-flavonoids. Stilbenes (resveratrol) and flavan-3-ols (catechin and epicatechin) were the main phenolic representative, resveratrol concentrations varying significantly between red-black (419.01–425.60 µg/g d.w.) and white (282.19 ± 4.14 µg/g d.w.) grape cultivars. Purified extracts (C-18 cartridge) exhibited higher antioxidant and antiradical activities compared to the crude extracts, and a higher antimicrobial effect, especially against Gram-positive (*Staphylococcus aureus*) pathogenic bacteria. Furthermore, *Saccharomyces cerevisiae* and *Oenococcus oeni* showed a reduced susceptibility even at high extract concentrations (>100 mg/mL). These findings indicate that grapevine canes represent a valuable source of natural bioactive compounds, that are currently insufficiently known and not exploited to their true functional and economic potential.

**Keywords:** antimicrobial potential; bioactive compounds; crude extracts; grapevine pruning; phenolic compounds purification; sustainable viticulture

## 1. Introduction

Grapevine (*Vitis* sp.) has played an important role in human history, being used for millennia for ceremonial, ornamental, dietary and medicinal purposes [1]. Nowadays, considering the weight of the edible portion, grapes are the first most produced fruit crop in the world [2]. The global vineyard surface area is estimated to be 7.3 mha, the European Union (EU) 3.3 mha, and the main purpose being wine production [3]. Romania is one of the world's largest wine producers (4.5 mhL) and fifth-largest among European wine-producing countries, with a vineyard surface area of 189.000 ha [3]. Pruning is employed to manage grapevine size and shape, to obtain maximum yields of high-quality grapes and to allow adequate vegetative growth for the following season. Grapevine is pruned during the dormant season, usually in late winter, the canes being the main byproducts, with a huge amount of discarded wood, respectively, at around 2–5 tons/ha each year [4,5]. By definition, grapevine canes are shoots that have reached about a year in age, that developed a visible brownish bark layer and lost the related leaves. Referring to the cultivated area and the quantity of canes resulting per hectare from the spring pruning, Romania generates almost 0.7 million tons of woody viticultural wastes every year.

Currently, grapevine canes exhibit a very low economic value, being burned (used locally as firewood) or chopped and incorporated into the soil, as compost on the field (lignin, cellulose, nitrogen, and potassium) or as soil mulch [4,6]. Recently, for decongesting the plots, local winemakers choose to collect and formally capitalize grapevine canes, to the detriment of their soil integration, the main reasons being related to the avoidance of disease perpetuation, limitation of changes in soil reaction (due to increase in acidity), obtaining of thermal energy (pellet production), reduction in expensive labor costs and compliance with environmental protection legislation. In the context of the worldwide "zero-waste economy" concept, various options of valorization were experimentally developed, grapevine canes being tested as an energy source, in the production of biorefinery compounds or bioactive carbon, hybrid particleboards (high lignin content) [6], being recently proposed as viticultural biostimulants [7], insecticide [8], wine preservative [9] or as an alternative to oak chips to improve the sensorial profile of wines [10]. However, most of these applications are not yet extended to large production, grapevine canes still representing an environmental problem. Also, it is not easy to diminish grapevine waste accumulation, as most of the practices are still limited by physical infrastructures or human resources [11]. On the other hand, a growing demand for natural products in consumer society has emerged in the past decade, plant biomass being intensively studied due to its richness in high-value compounds [12]. Polyphenolic compounds represent one of the most important classes of secondary metabolites in *V. vinifera* L. plants, exhibiting a large spectrum of pharmacological and therapeutic benefits [13–16]. In grapevine canes, both flavonoids and non-flavonoids were identified, mainly as proanthocyanidins and stilbenes, respectively [17]. Liquid chromatography (HPLC) with photodiode array detection (PAD) is a very popular and useful tool for the analysis of phenolic compounds from plant extracts [18,19]. Using liquid chromatography, various compounds from the stilbene group, mainly resveratrol and viniferins were identified in grapevine canes [20–22]. However, up to 41 stilbenes have been found in grapevine canes [4]. Phenolic acids, such as protocatechuic, vanillic, syringic, caffeic, ferulic, coumaric and sinapic, were also identified and quantified in grapevine cane alcoholic extracts [23–25]. Considering the high level of waste produced by pruning in worldwide viticulture, Rayne et al. [26] estimated that the extraction of stilbene from grape canes may reach a global economic value of over $30 billion.

The extraction of bioactive compounds from plant materials is the first and one of the most important steps in the preparation of dietary supplements or nutraceuticals, food ingredients and pharmaceutical products [27]. As already known, phenolic compounds are unstable, each plant source demanding an individual approach for extraction. Due to the diversity of the polyphenol classes, it is very difficult to identify a generally applicable extraction procedure. Thus, various solvents were proposed for the extraction of useful

compounds from grapevine canes (especially stilbenes), such as water, methanol, ethanol, acetone [4,28–31] or sodium hydroxide [23]. In addition, solvent concentration and the ratio between various solvents may vary widely depending on the grapevine species, cultivar and growing area. Several extraction procedures such as maceration at laboratory temperature, extraction at elevated temperature, fluidized-bed extraction, Soxhlet extraction, microwave-assisted extraction, accelerated solvent extraction [20,32], superheated liquid extraction, microwave-assisted extraction, ultrasound-assisted extraction [24], supercritical fluid extraction, pressurized liquid extraction [33] or subcritical-water extraction [31] were tested for obtaining the polyphenol fraction of grapevine canes. By definition, an extraction technique must ensure the complete extraction of the useful compounds, not be laborious, energy or time-consuming, be cost-effective, not require large volumes of solvents, be selective and avoid enzymatic or non-enzymatic degradation of the bioactive compounds. Also, the solvent used must be selective, with a boiling point as low as possible, and not be corrosive or toxic. According to Zwingelstein et al. [12], the emerging methods are not always more efficient than the classical one, all the protocols requiring important volumes of solvents. Also, new laboratory methods require expensive investments, being generally less time-consuming. Between the different solvents used, ethanol is preferred when the destination of the extracts is food or pharmaceuticals to avoid the high toxicity of methanol, acetone, or acetonitrile solutions, although the concentrations of useful compounds can be slightly lower [27,34]. Under these conditions, solid–liquid extraction using aqueous ethanol solutions as the solvent remains one of the main options for obtaining polyphenolic compounds when considering the extraction yield, low toxicity, accessibility and large-scale practicality, thus fulfilling the required basic extraction conditions to the greatest extent.

Purification of polyphenolic extracts is often necessary, as the solvent systems are not sufficiently selective. Considerable amounts of accompanying compounds may be extracted and concentrated in the crude extracts, which can influence the stability, biological activity and analysis of the individual phenolic compounds [35]. Polyphenol purification using solid-phase extraction (SPE) enables the removal of the interfering compounds present in the crude extracts [36]. Mini-columns containing C-18 chains bonded to silica retain hydrophobic organic compounds (phenolics), while allowing matrix interference such as sugars or acids to pass through to waste. The isolation and purification step using SPE makes the efficient analysis of individual polyphenolics possible [37].

Data regarding the biological potential of the purified grapevine cane polyphenolic extracts are very scarce, most of the research being mainly directed at the identification and quantification of stilbenes, rather than to a synergistic valorization of the plant material composition. Recent studies have shown that grapevine cane crude extracts are rich in bioactive compounds with good antioxidant potential [38,39] and particular antimicrobial activity against different bacteria and yeasts [39,40], but the reported results vary widely depending on the grapevine genotype and the tested microorganisms, in generally being directly correlated with the phenolic compounds' concentration [4].

Since a high amount of woody waste is produced annually by grapevine pruning, the aim of this study was to assess the physico-chemical composition of canes of three *Vitis vinifera* L. cultivars in high demand for planting in temperate climate vineyards, to determine the main conditions for the extraction of phenolic compounds, to evaluate the phenolic composition and the antioxidant, antiradical and antimicrobial activities of the crude and purified (C-18 SPE cartridge) extracts. The antimicrobial properties of the polyphenolic extracts were tested on both Gram-positive and Gram-negative pathogenic species, as well as on species with high technological importance in winemaking (yeast and lactic acid bacteria). Knowing the type and amounts of useful compounds available in grapevine canes, as well as their biological properties, offers new possibilities for the valorization of industrial waste and a better appreciation of their economic and functional value, as a potential source of high-value phytochemicals.

## 2. Materials and Methods

### 2.1. General Climate Conditions of the Vineyard

Copou-Iasi viticultural center is located in the Iasi vineyard, in the northeast of Romania, 47°10′ northern latitude and 27°35′ eastern longitude, with favorable eco-pedoclimate conditions for growing grapevines. The climate is temperate continental, with excessive nuances, characterized by large contrasts between seasons, with harsh and dry winters, hot summers often with droughts [41]. The plot was planted at an altitude of 184 m, on a slight slope (3%) with a southern exposition (orientation of rows N–S), in a cambic chernozem soil with a clay–loamy texture, 6.8 pH units, 2.7% humus content, formed on marls with sand insertions, with the phreatic water depth at over 3 m [42]. Meteorological data were collected daily by means of a weather station located in the experimental plot, using the AgroExpert® 1.6 software. The climate analysis of the last 20 years indicates an average annual air temperature of 10.5 °C, with a maximum value of 11.6 °C in 2019 and a minimum of 9.5 °C in 2001. In 2018, the absolute minimum air temperature was −19.7 °C (24 January 2018) and the absolute maximum air temperature was 32.0 °C (5 August 2018), the annual amount of precipitation was 727.8 mm, of which 460 mm was during the growing season. In 2019, the absolute minimum air temperature registered between January and March (until spring pruning) was −12.0 °C (8 January 2019).

### 2.2. Biological Material

This study has been carried out on 1-year-old grapevine canes with a minimum of ten buds, harvested manually during the 2019 spring pruning (1–5 March) from three *Vitis vinifera* L. cultivars: Sauvignon Blanc (white grapes; VIVC no. 10790), Pinot Gris (rose grapes; VIVC no. 9275) and Cabernet Sauvignon (black grapes; VIVC no. 1929) [43], growing in the Iasi vineyard, Copou viticultural center. Grapevines were 20 years old, grouped into plots with planting distances 2.2 m between rows and 1.2 m between plants (≈3700 plants/ha), with an average load of 35 to 40 buds/vine stock. Plants were grafted on the hybrid rootstock Kober 5 BB (*Vitis berlandieri* Planch. × *Vitis riparia* Michx.), the technological operations applied being specific to the industrial vineyards. Plants were not irrigated or fertilized. Visually clean and healthy grapevine canes were cut into 25 cm pieces, dried in the dark at room temperature (21 ± 2 °C) for one week and then in a drying oven (UF55, Memmert GmbH, Schwabach, Germany) at 50 °C until constant weight [40], and subsequently cut into cylinders of 0.5–0.8 cm, ground using an electrical grinder (GT110838, Tefal, Rumilly, France), passed through a 0.5 mm sieve (particle size < 0.5 mm) and stored in sealed bags at room temperature until extraction (Figure 1a).

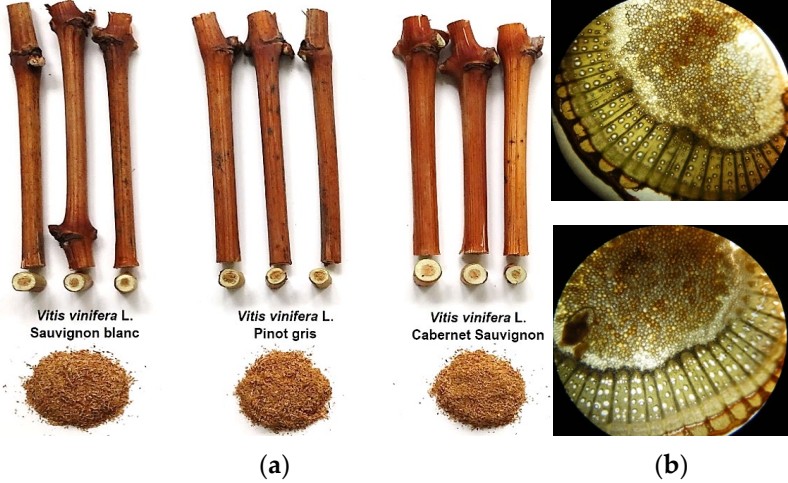

(**a**)                                                                 (**b**)

**Figure 1.** The biological material analyzed and aspects of processing (**a**), section under the microscope—the dark areas in the medullary zone (pith) indicate the starch content stained with Lugol's iodine solution (**b**).

### 2.3. Physico-Chemical Determinations

Grapevine canes' diameters and the length of the cylinders were determined using a digital vernier caliper (Powerfix, Neckarsulm, Germany) ($\pm$0.01 mm) and an ATX 224 (Shimadzu, Kyoto, Japan) analytical balance was used for the weighing. Moisture content (%) was determined by drying the plant material at 105 °C until constant weight (drying oven UF55, Memmert GmbH, Schwabach, Germany). Total mineral content was assessed gravimetrically (%), by heating the plant sample in a muffle furnace for 6 h at 525 °C, after a preliminary flame burning.

Total carbohydrate content was determined by extraction of total sugars (80% ethyl alcohol) and starch (52% perchloric acid) from dried (65 °C) ground grapevine canes, reaction with 0.2% anthrone in 99% sulfuric acid and spectrophotometric measurement of the color intensity at 580 nm (% glucose) according to procedure presented in STAS 220/10–85 [44]. Total carbohydrates (total sugars + starch) were expressed as a percentage (g/100 g dry weight).

To highlight the presence of starch, five internodal slides obtained by a penknife were treated for 3 min with a drop of Lugol solution (10 g/L potassium iodide and 2.5 g/L iodine, in distilled water) for starch–iodine complexation (Figure 1b) and observed under a microscope [45]. The Biuret test was used to approximate the amount of protein and amino acids present in the extracts. Thus, 3 mL of extract, 2 mL of 4% sodium hydroxide and a few drops of 1% copper sulphate solution were vortexed in a test tube. The presence of proteins was indicated by the formation of a pink to violet color [46].

### 2.4. Extraction Procedures

For the extraction of the phenolic compounds, 5 g amounts of powdered grapevine cane (in triplicate) were transferred into 250 cm$^3$ Erlenmeyer flasks and extracted with 100 mL of solvent containing various concentrations of ethanol (0, 40, 50, 60, 70, 80, 90 and 95% v/v) (plant material/solvent ratio of 1:20 w/v). Experimental extractions were performed initially at 25 °C, 12 h, in the dark, in static conditions. After centrifugation at 7500 rpm for 10 min (CENHBN-600, MRC, Holon, Israel) the supernatant was collected to form the crude extract for each variant. After assessing the optimal solvent concentration, the time (2, 4, 6, 8, 12, 16 and 24 h), temperature (20, 25, 30, 35, 40, 45, 50, 55 and 60 °C) and the stirring conditions (8 h intermittently—20 s every hour, and continuously at 150 rpm for 2, 4, 6 and 8 h) were determined to maximize phenolic compound extraction. A laboratory digital thermostatic bath (model 601/5, Nahita, Japan) and an orbital platform shaker (model 3006, GFL, Burgwedel, Germany) were used to perform the experiments. The application of ultrasound (ultrasonic bath EMMI-04D, 42 KHz, Emag, Mörfelden-Walldorf, Germany) at constant temperature (25 °C) was also tested (3, 6, 9, 12 and 15 min). To obtain the main extracts, the best experimental conditions were applied, with three extraction phases being performed for each sample. The grapevine cane crude extracts (CE) were stored at −20 °C, in dark colored glass containers with screw caps, until analysis.

### 2.5. Chlorophyll and Carotenoid Content

Chlorophyll (a and b) and total carotenoid content in the grapevine cane crude extracts (with 96% ethanol) were determined by spectrophotometry, according to the protocol and formulas presented by Lichtenthaler and Buschmann [47]. The absorbance of the samples was recorded at 470, 662, 645, and 710 nm as part of a full scan (400–800 nm) of the samples using a Specord 200 plus spectrophotometer (Analytik Jena, Jena, Germany), and the quantitative results were expressed as µg/g dry weight.

### 2.6. Total Phenolic and Flavonoid Content

Total polyphenolic content (TPC) was determined using Folin–Ciocalteu reagent (Scharlab, Barcelona, Spain), according to the method proposed by Singleton and Rossi [48]. Gallic acid (g/100 g d.w.) was used as a standard (y = 1.2114x + 0.0118; $R^2$ = 0.9910). Flavonoids (Fl) were precipitated with 10 mL of formaldehyde (8 mg/L, in distilled water)

at pH < 0.8 (made with 5 mL of HCl 50/50 *v/v*) as recommended by Tibiri et al. [49]. The mixtures were vortexed (Classic vortex F202A, Fisherbrand, Italy), and after 24 h at room temperature, flavonoids were separated by centrifugation (3000 rpm, 10 min); the supernatant containing all phenolic compounds except flavonoids was collected. Absorbance was measured in the same way as for the total phenolics, using a Specord 200 plus computer-controlled double-beam UV–Vis spectrophotometer (Analytik Jena, Germany). The non-flavonoid (NFl) content was calculated as (X–Y), where X is the total phenolic content and Y the flavonoid content. Also, the global assessment of the polyphenolic content, as total polyphenolic index (TPI), was performed by measuring the absorbance of the extracts at 280 nm, after the corresponding dilutions [50].

### 2.7. Polyphenolic Extract Purification

After determining the extraction conditions, a part of the crude extract (100 mL) was concentrated to dryness in a rotary vacuum evaporator (HS-2005V, Hahn Shin Scientific, South Korea) at 34 °C. The solid was dissolved in 50 mL 0.01% HCl (*v/v*) in distilled water and used for further purification. The polyphenolic aqueous solutions were passed through a preconditioned C-18 SPE Chromabond® cartridge (Macherey-Nagel, Düren, Germany), with a 6 mL volume, 500 mg sorbent, silica base material, pore size 60 Å, particle size 45 μm, with a pH stability of 2 to 8. Phenolic compounds were adsorbed onto the C-18 mini-column, while sugars, acids and other water-soluble compounds were removed by washing the cartridge with 2 volumes of 0.01% aqueous HCl (*v/v*). The percolation was carried out slowly and constantly (4 mL/min). The purified polyphenolic extracts (PE) were obtained by the elution of the cartridge sorbent with 25 mL of 96% ethanol and used for HPLC analysis.

### 2.8. HPLC Procedure

Separation and quantification of individual phenolic compounds in grapevine cane purified ethanolic extracts were performed by means of a Waters 2695e Alliance high-performance liquid chromatography (HPLC) system, coupled with a 2998 diode array detector (PDA Detector), controlled by Empower® 3 software (Waters, Milford, MA, USA) with a capillary C-18 column Waters XBridge (50 × 4.6 mm, 3.5μm), thermostatically kept at 30 °C. All samples were filtered through 0.45 μm syringe filters. A total of 20 μL of sample was injected for determination. The mobile phase A was represented by a solution of 0.1% trifluoroacetic acid (TFA) in water, while for solvent B a solution of 0.1% TFA in acetonitrile was used, with a flow rate of 0.7 mL/min. All solvents were of HPLC grade (Merck, Darmstadt, Germany). The chromatograms were monitored at 280 nm, while compound identification was based on available standards. Individual phenolic compound quantification was performed using standard curves of external standards ($r^2 > 0.99$), by plotting the peak areas against the concentrations (μg/mL).

### 2.9. DPPH Scavenging Activity

2,2-diphenyl-1-picrylhydrazyl (DPPH) free radical scavenging activity of crude and purified extracts was measured according to the procedure proposed by Brand-Williams et al. [51]. The reaction mixture contained 100 μL of extract (0–100 μg/mL) and 3.9 mL of 2.36% DPPH in 96% ethanol solution. The test tubes with the mixture were incubated at 37 °C for 30 min. The reduction in absorbance was measured at 517 nm. The antiradical activity was reported as a percentage of scavenged DPPH* (%) = $((\text{Abs}_{control} - \text{Abs}_{test})/\text{Abs}_{control}) \times 100$. Half maximal inhibitory concentration (IC50), as the concentration of extract necessary to decrease the initial concentration of DPPH by 50% was calculated. Ascorbic acid and gallic acid (0–100 μg/mL) were used as positive controls.

### 2.10. Phosphomolybdenum Assay

The phosphomolybdenum assay was used to evaluate the total antioxidant capacity of the extracts. An aliquot of 0.1 mL of extract was mixed with 1.0 mL of the reagent

solution prepared by dissolving 0.6 M sulfuric acid, 28 mM sodium phosphate and 4 mM ammonium molybdate and incubated at 95 °C for 90 min. The test tubes were cooled to room temperature and the absorbance was measured at 695 nm against reagent blank [52]. Ascorbic acid and gallic acid (0–100 µg/mL) were used as positive controls.

### 2.11. Ferric Reducing Power

The reducing capacity of the extracts was performed by a $Fe^{3+}$ to $Fe^{2+}$ reduction assay, according to the protocol described by Oyaizu [53]. The reaction mixtures containing 2.5 mL of 0.2 M phosphate buffer (pH 6.6), 2.5 mL of potassium ferricyanide (1% *w/v*), and 1 mL of extract (0–100 µg/mL) were incubated in a water thermostatic bath (601/5, Nahita, Japan) at 50 °C for 20 min. After cooling and addition of 2.5 mL of trichloroacetic acid (10%, *w/v*), the mixtures were centrifuged at 3000 rpm for 10 min. A total of 2.5 mL of supernatant was mixed with 2.5 mL of distilled water and 0.5 mL of ferric chloride (0.1%, *w/v*) and the absorbance was measured at 700 nm against a reagent blank solution. The IC50 value (µg extract/mL) was the effective concentration of extract at which the absorbance was 0.5 for the reducing power and was obtained by interpolation from the linear regression analysis [38]. Ascorbic acid and gallic acid (0–100 µg/mL) were used as positive controls.

### 2.12. Phenanthroline Assay

The reducing capacity of grapevine cane extracts was also evaluated by the phenanthroline assay. A total of 0.6 mL of sample (0–100 µg/mL), 1 mL of 0.2% ferric chloride solution (in methanol) and 0.5 mL of 0.5% 1,10-phenanthroline solution (in methanol) were placed into a 10 $cm^3$ volumetric flask and made up to volume with methanol. The obtained solution was mixed and incubated in the dark for 20 min at 30 °C. The absorbance of the orange-red solutions was measured at 510 nm against a reagent blank (the sample volume was replaced with methanol) [54]. Ascorbic acid and gallic acid (0–100 µg/mL) were used as positive controls.

### 2.13. Antimicrobial Activity Assay

Antimicrobial activity of grapevine cane extracts was evaluated by the disk diffusion method on agar medium. The pathogenic bacterial strains of Gram-positive bacteria, *Staphylococcus aureus* (ATCC-25923) and Gram-negative bacteria, *Escherichia coli* (ATCC-25922), were cultured in Mueller–Hinton broth (Merck, Darmstadt, Germany) at 37 °C. The bacterial suspension was adjusted to 0.5 McFarland standard by turbidity ($1.5 \times 10^8$ CFU/mL). The standardized bacterial inoculum was spread into Petri dishes (Ø 90 mm) containing 15 mL sterilized Mueller–Hinton agar medium (121 °C for 15 min). Then, aseptically dried filter paper discs (6 mm in diameter) containing the tested extracts in 5% DMSO (0–100 mg/mL), were placed on the agar surface. The Petri dishes were incubated at 37 °C, 24 h, and the diameters of inhibition growth zones were measured (digital vernier caliper). Discs loaded with 5% DMSO were used as negative control, while gentamicin (10 µg/mL) was the positive control. Tests were performed in triplicate and the antibacterial activity was expressed as the mean zone of inhibition diameters (mm) produced by the various concentrations of extract. Minimum inhibitory concentration (MIC) and minimum bactericidal concentration (MBC) were determined in Mueller–Hinton broth. Test tubes with sterile medium containing the extracts and the test bacteria ($10^6$ CFU/mL) were incubated at 37 °C under aerobic conditions. After 24 h, 1 mL of homogenized media was spread into Petri dishes with Mueller–Hinton agar medium and incubated at 37 °C. After 24 h, MIC was determined as the lowest concentration of the extract (mg/mL) that inhibits the visible growth of the microorganisms, while MBC was determined as the lowest concentration of extract that prevents the growth of bacteria [55]. MBC/MIC ratio was calculated according to Mogana et al. [56], in order to evaluate the bactericidal (MBC/MIC $\leq$ 4) or bacteriostatic (MBC/MIC > 4) activity of the extracts on tested microorganisms.

Through similar procedures, the susceptibility to extracts of some microorganisms used in winemaking was tested, respectively, a commonly used yeast strain (*Saccharomyces*

*cerevisiae*, Vinoferm Aroma®, Essedielle, Villa Caldari, CH, Italy) and a lactic acid bacteria strain (*Oenococcus oeni*, Viniflora Oenos®, Chr. Hansen, Hørsholm, Denmark). The yeast ($10^6$ CFU/mL) was tested on YEPD medium (Scharlau, Barcelona, Spain) (48 h, 25 °C), while the lactic acid bacteria strain ($10^8$ CFU/mL) was cultured on MRS medium [57], 48 h, 30 °C, in anaerobiosis (GENbag anaerobic, BioMérieux, Craponne, France).

### 2.14. Statistical Procedures

Data were reported as the mean of three replicates, having specified the standard deviation (±). Analysis of variance (ANOVA test) was initiated to investigate significant differences between data in XLSTAT®: Statistical software (2023 edition), within Microsoft® Excel 2019 software. The method used to discriminate among the means was Duncan's multiple range test at 95% confidence level. *p* values lower than 0.05 ($p < 0.05$) were considered significant. Different letters indicate significant differences between data. Regression analysis was performed to look for relationships between the experimental results.

### 3. Results and Discussion

### 3.1. Physico-Chemical Characterization of Grapevine Canes

*V. vinifera* L. canes, manually harvested during the spring pruning, did not vary significantly in terms of diameter (0.68–0.81 cm). A significant difference in moisture and dry matter content was observed. Canes of Cabernet Sauvignon (CS) cultivar showed a higher moisture (46.10%), while canes of the Sauvignon Blanc (SB) cultivar showed a higher total dry matter content (58.20%) (Table 1).

**Table 1.** Physico-chemical characteristics of grapevine canes at pruning.

| Features | Cultivar | | |
|---|---|---|---|
| | **Sauvignon Blanc** | **Pinot Gris** | **Cabernet Sauvignon** |
| Diameter (cm) | 0.68 ± 0.11 a | 0.74 ± 0.08 a | 0.81 ± 0.10 a |
| Moisture (%) | 41.80 ± 0.88 b | 44.40 ± 0.91 a | 46.10 ± 0.74 a |
| Total dry matter (%) | 58.20 ± 0.88 b | 55.60 ± 0.91 a | 53.90 ± 0.74 a |
| Starch (%) | 5.60 ± 0.28 a | 6.32 ± 0.41 a | 5.54 ± 0.32 a |
| Sugars (%) | 6.43 ± 0.22 b | 7.66 ± 0.28 a | 7.08 ± 0.32 a |
| Total carbohydrates (%) | 12.03 ± 0.50 b | 13.98 ± 0.69 a | 12.62 ± 0.64 ab |
| Minerals (%) | 4.02 ± 0.16 a | 3.94 ± 0.11 a | 3.73 ± 0.20 a |

Note: Values are presented as the mean of a minimum of three replicates, with standard deviations (±). Different letters within the same row indicate significant differences ($p < 0.05$) in Duncan's multiple range test.

The carbohydrate amount stored in grapevine canes is an indication of the health and vigor of the previous season's growth [58]. Also, the carbohydrate content is an indicator of wood maturity and quality, as the initial growth of emerging shoots in spring is mainly at the expense of carbohydrates. Total carbohydrate content, as the sum of starch and sugar amounts, varied significantly depending on genotype, from 12.03 to 13.98%. The highest starch and sugar contents were determined in the canes of Pinot Gris (PG) cultivar, of 6.32 and 7.66%, respectively. Çetin et al. [59] reported higher concentrations of carbohydrates in the canes of some grape varieties from Turkey, up to 44%, several factors being generally involved in the variation of carbohydrate concentrations (species, cultivar, rootstock, climate, mineral nutrition, trellising system or crop level). However, according to the Romanian standard 220/10-85, the canes with a total content of carbohydrates (starch and sugars) lower than 12.00% cannot be used in grafting for vegetative propagation [44]. The degree of cane maturation, defined by a certain optimal level of reserve substances, determines the frost resistance of plants. The presence of starch in the cane tissues was visually highlighted by treating the sectioned surface with iodine solution and microscopic observation of the color change. Starch accumulates mainly in the inner layers of the xylem. With the decrease in temperatures, the reduction in the starch content is manifested through the hydrolysis reaction and the amount of sugar increases accordingly [60]. The presence

of starch was better observed in the PG variety, the stained section being wider and better defined (Figure 1b), in direct correlation with the chemically determined concentrations.

Regarding the total mineral content, the values varied non-significantly among cultivars (3.73–4.02%).

### 3.2. Determination of the Extraction Parameters

Solid–liquid extraction (SLE) is based on diffusion and osmosis processes and is performed by maceration [61]. SLE is frequently applied due to some important advantages: simple operation and apparatus required, low cost, possibility of applying various temperatures, stirring or ultrasound that facilitates the solubility and penetration of the solvent [62]. Some disadvantages such as low selectivity, the use of large solvent volumes, or the necessity of repeated extractions are also mentioned. Accordingly, solvent selectivity can be solved by purifying the crude extract, while the number of repeated extractions can be minimized by increasing the initial solid-to-solvent ratio. Moreover, the use of large volumes of solvent is counteracted by the vacuum concentration of the final extract and solvent recovery. The extraction yield depends mainly on the type of solvent, extraction time and temperature, and sample-to-solvent ratio [14]. In order to maximize the polyphenol amount extracted, the main influencing factors of the process were tested (solvent concentration, temperature, contact time, stirring conditions and ultrasounds application). Dry PG canes were used for testing the extraction conditions, at a sample-to-solvent ratio of 1:20 (*w/v*), as established in preliminary tests (data not shown). Also, previous studies recommended a similar solid–liquid ratio [59], although several solvent-to-solid ratios have been proposed. According to the mass transfer principles, an increase in the hydromodule (solvent-to-solid ratio) improves the diffusion rate in a solid–liquid extraction [63].

### 3.2.1. Solvent Concentration

Polyphenols are most soluble in organic solvents less polar than water [36]. Thus, the efficiency of the hydro-alcoholic mixtures is explained by their polarity and their ability to form hydrogen bonds with phenolic compounds [12]. The choice of the extraction system was made based on the analyte solubility, the balance of cost, safety and environmental concerns as previously mentioned by Naviglio et al. [61]. Ethanol is recommended as an efficient solvent for polyphenol extraction and is generally recognized as safe (GRAS) for potential application of the extracts in the food or drug fields [64].

Triplicate static solid–liquid extractions with ethanol:water mixtures were applied for testing the solvent extraction capacity. The efficiency of the extraction systems was evaluated by the total amount of phenolic compounds extracted from the dry plant material under similar experimental conditions. A significant increase in the phenolic compound's concentration was determined up to an ethanol concentration of 70% (*v/v*) (19.88 mg GAE/g d.w.) (Figure 2a).

Increasing the ethanol concentration no longer corresponded to a significant increase in the total amount of phenolic compounds extracted (+6.8%). Thus, 70% (*v/v*) ethanol/water solution was about five times more effective than water in the extraction of phenolic compounds from grapevine canes, under similar laboratory conditions.

Previous studies have reported various ethanol concentrations that were effective in the extraction of phenolic compounds from grapevine canes. A 50:50 (*v/v*) ethanol/water solvent was used by Moreira et al. [40], 60:40 (*v:v*) ethanol:water mixture was proposed by Çetin et al. [59] and Houillé et al. [65], while Ewald et al. [66] reported that an ethanol/water 80:20 (*v/v*) solution was effective for stilbene extraction (from canes dried in the dark, for up to 6 months). The use of a 70% ethanol in water concentration (*v/v*) was also reported [13].

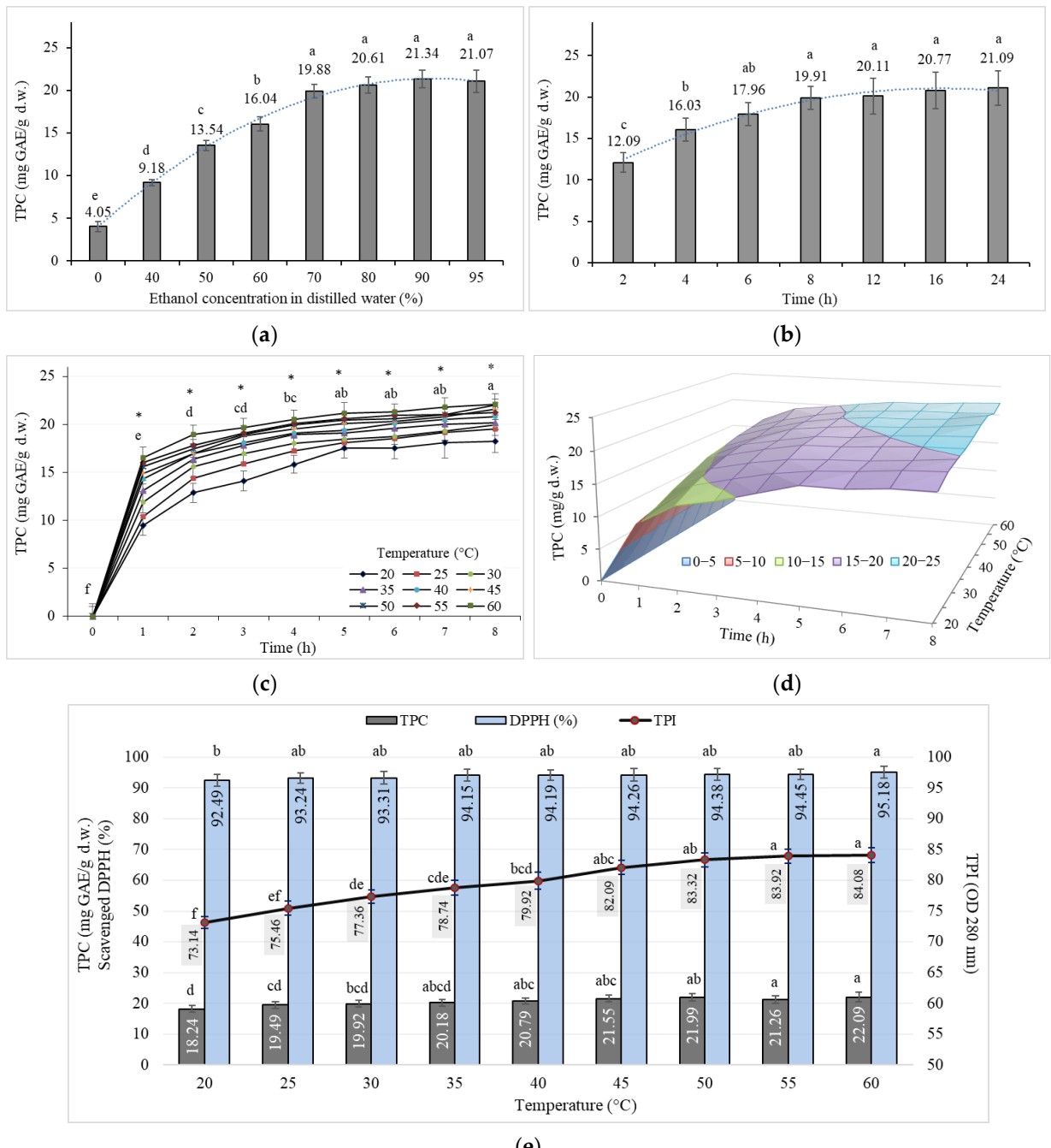

(**a**)

(**b**)

(**c**)

(**d**)

(**e**)

**Figure 2.** The influence of ethanol concentrations (0 to 95% *v/v*, 25 °C, 12 h) (**a**), time of contact (2 to 24 h, in 70% ethanol) (**b**) and temperature (20 to 60 °C, in 70% ethanol) (**c**) on the total concentration of phenolic compounds extracted from grapevine canes (Pinot Gris); the three-dimensional surface chart plot (temperature, time and phenolic compounds concentration) (**d**) and the relationship between the total phenolic compounds' content and antioxidant activity (DPPH%) at different extraction temperatures (**e**). Note: TPC—total polyphenolic content; TPI—total polyphenolic index, evaluated as optical density at 280 nm; GAE—gallic acid equivalent; d.w.—dry weight. Experimental extractions were performed in the dark, in static conditions, as a single stage extraction (sample-to-solvent ratio 1:20 *w/v*). Values are presented as mean results of three independent experiments, with standard deviations (error bars). * indicates significant differences (*p* < 0.05; ANOVA test) between mean amounts of phenolic compounds extracted at various temperatures (on vertical). Different letters indicate significant differences between the mean values in Duncan's multiple range test (on horizontal). The three-dimensional surface chart plot was performed in Microsoft® Excel software.

### 3.2.2. Extraction Time

After determining the optimal solvent concentration, the time required for the extraction using 70% (*v/v*) ethanol as solvent was assessed. Thus, in the first four hours of maceration (1:20 *w/v* ratio), in static conditions, at 25 °C, over 80% (16.03 mg GAE/g d.w.) of the phenolic compounds obtained after eight hours was extracted (19.91 mg GAE/g d.w.) (Figure 2b). However, after eight hours of extraction the concentrations of phenolic compounds increased non-significantly, the values obtained after 24 h of maceration being higher by about 5.50% (21.09 mg GAE/g d.w.). It was also observed that an extension of the extraction time, corresponded to a higher variation in the extracted phenolic compounds' concentrations (showed as standard deviation).

### 3.2.3. Temperature

There are only few studies published on the impact of temperature on phenolic compound extraction from grapevine canes using conventional methods. Theoretically, under high temperatures, plant tissues are softened and the phenolic compounds can be more easily extracted into the solvent [67]. At higher temperature, ethanol showed a higher capacity to solubilize phenolic compounds, with surface tension and solvent viscosity decreasing with temperature, and improving sample penetration [68]. However, the ethanol/water mixtures cannot afford to use higher temperatures because the alcohol boiling point is below 80 °C [12]. The kinetics of the phenolic compounds' extraction by applying different temperatures is shown in Figure 2c. As expected, increasing the temperature contributed to a higher extraction yield. Regardless of the applied temperature, the highest amounts of phenolic compounds were obtained in the first two hours of extraction (70 to 85%). After this point, the concentrations of phenolic compounds did not increase significantly with the increase in the extraction time.

The three-dimensional surface chart plot indicated the direct relationship between temperature, time and the amounts of phenolic compounds extracted (Figure 2d). Thus, a high extraction yield was achieved in the first hours of the process, at a temperature starting at 35 °C. There is a lack of data regarding the solvent—time—temperature interactions. According to Naczk and Shahidi [69], a prolonged extraction time at high temperature may even decrease the extraction yield due to the oxidation and degradation of some phenolic compounds. After eight hours, the total phenolic compounds content (TPC) varied between 18.24 mg GAE/g d.w., at 20 °C, and 22.09 mg GAE/g d.w., at 60 °C (Figure 2e). For precise observation, in this case, the total phenolic content was estimated both by using the Folin–Ciocalteu reaction (F–C) and by measuring the absorbance of the extracts at 280 nm. The F–C reaction is the simplest economic technique for the measurement of phenolic compounds in foods, herbs, and other plant extracts, and generally provides accurate data for several groups of phenolic compounds, but is often associated with some limitations (e.g., participation of protein–tannin complexes or various nitrogen compounds) [50,70–72]. Total polyphenolic index (TPI), determined by measuring absorption at 280 nm, seems preferable to the F–C test as it presents a number of advantages, including speed and reproducibility [50]. TPI is based on the characteristic absorption of the benzene cycles of the majority of phenols in the UV spectrum, at 280 nm. However, certain molecules, such as cinnamic acids and chalcones, have no absorption maximum at this wavelength, for this reason, when the conclusion is not clear, the two determinations must be performed in parallel. TPI values of grapevine canes were highly correlated with TPC values ($r^2$ = 0.9438), but a better correlation was identified between TPI and DPPH scavenging activity of the extracts ($r^2$ = 0.8816), than between TPC and DPPH scavenging activity ($r^2$ = 0.8782). DPPH scavenging activity varied significantly between the extracts obtained at 20 °C and at 60 °C, higher values being obtained when the extraction was performed at higher temperatures (Figure 2e). However, between these two extreme temperatures the DPPH scavenging activity did not increase significantly with the increase in the phenolic compounds' concentration.



### 3.2.4. Stirring Conditions

The extraction of the phenolic compounds from the ground dry plant material was improved by introducing a stirring/homogenization step (intermittent or continuous for 2, 4 or 8 h), performed using an orbital platform shaker, at 150 rpm (at 25 °C). After the stirring period, the samples were kept in static conditions until the end of the initially established time (eight hours), in order to compare the results. The best yield of phenolic compounds was obtained by intermittent stirring every hour for 20 s (21.61 ± 0.41 mg GAE/g d.w.) and also by continuous homogenization for eight hours (21.84 ± 0.52 mg GAE/g d.w.), ensuring a significant increase of about 9% compared to the stationary extraction (19.84 ± 0.58 mg GAE/g d.w.) (Figure 3a). Considering the energy consumption and obtained yields, intermittent stirring of the samples during the extraction process (20 s, every hour) or the continuous stirring of the samples (150 rpm) for a maximum of four hours may be recommended. The increase in the extraction time under continuous stirring lead to non-significant increases in the obtained yields.

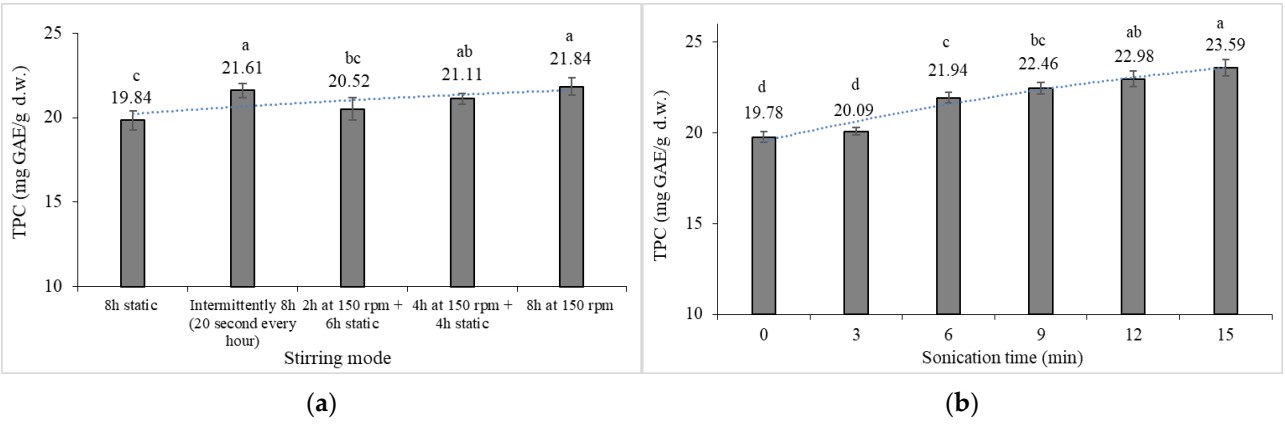

**(a)**                                                                                                   **(b)**

**Figure 3.** The influence of stirring conditions (**a**) and ultrasound application (**b**) on the total concentration of phenolic compounds extracted from grapevine canes (Pinot Gris). Note: TPC—total polyphenolic content; GAE—gallic acid equivalent; d.w.—dry weight. Experimental extractions were performed at 25 °C, as a single stage extraction (sample-to-solvent ratio 1:20 *w/v*). Values are presented as the mean of three independent experiments, with standard deviations (error bars). Different letters indicate significant differences in Duncan's multiple range test ($p \leq 0.05$).

### 3.2.5. Ultrasound Application

Ultrasound application is a simple, inexpensive, fast and efficient operation for extracting the bioactive compounds from plant materials [73]. The process is based on the use of sound waves, usually in a range of 20 to 100 kHz, that pass through a liquid medium, facilitating the extraction of bioactive compounds from the plant matrix [74]. Triplicate samples of dried ground grapevine canes in 70% (*v/v*) ethanol solution, in similar experimental conditions (25 °C; solid-to-solvent ratio 1:20 *w/v*; single stage extraction), were sonicated (42 KHz) for 3 to 15 min. The control was performed under conventional conditions (8 h, in static). The amount of total phenolic compounds (TPC) extracted from the plant material increased with the extension of sonication time, but from the statistical point of view, after 6 min of ultrasound application, the concentrations of useful compounds increased non-significantly (+6.9%) (Figure 3b). The most important increase in TPC was observed between minutes 3 and 6, when the values increased from 20.09 to 21.94 mg GAE/g d.w. (+8.4%). Comparing the two extraction processes, stirring and ultrasound application, it was observed that the intermittent homogenization of the extraction mixture for eight hours (20 s each hour, 150 rpm) was equivalent to 6 min of sample sonication at 42 kHz. One of the two processes can be used for the extraction of phenolic compounds from dried ground grapevine canes, depending on the requirements and infrastructure of the research laboratories.

### 3.3. Obtaining the Crude and Purified Polyphenolic Extracts

Considering the preliminary results, the extraction of the phenolic compounds from the canes of three *V. vinifera* L. cultivars (Sauvignon Blanc, Pinot Gris and Cabernet Sauvignon) was initiated. Thus, according to the obtained data, the ground plant material (10 g) was extracted with a 70% (*v/v*) ethanol aqueous solution (solid–liquid extraction), 4 h at 35 °C, after an ultrasound pretreatment of 6 min (42 kHz). Three extraction stages were performed, by recovering the solid after centrifugation and collection of the supernatants, the final ratio between plant material and solvent reached 1:20 (*w/v*). In the first stage about 72% of the total phenolic compounds were extracted (Figure 4a). The other two stages of extraction helped to exhaust the vegetal material in polyphenolic compounds, with a yield of about 6% being obtained in the last phase. A very low standard deviation (±1%) between the percentages obtained for the three different grapevine cultivars in each stage was highlighted, indicating the high repeatability and the possibility of using the method for different grapevine cultivars.

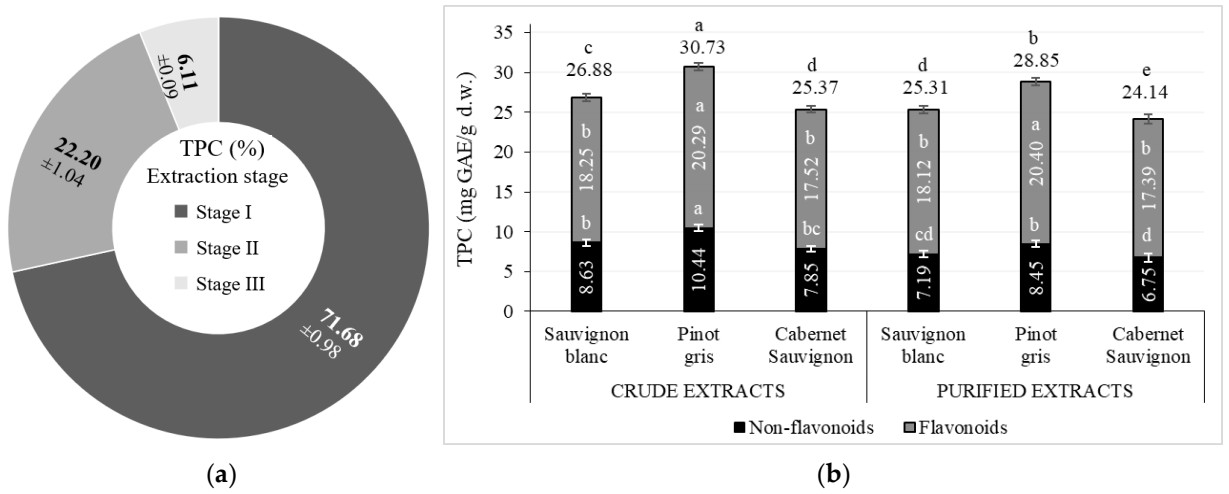

(**a**)  (**b**)

**Figure 4.** The yield of phenolic compounds (%) on each extraction phase (**a**), and the flavonoid and non-flavonoid content of the crude and purified grapevine cane extracts (**b**). Note: TPC—total polyphenolic content; GAE—gallic acid equivalent; d.w.—dry weight. Values are presented as the mean of independent experiments (*n* = 3). Error bars indicate standard deviations (±). Different letters indicate significant differences in Duncan's multiple range test ($p \leq 0.05$).

TPC of the cumulated fractions (crude extract) varied significantly depending on the cultivar, with values between 25.37 ± 0.56 (Cabernet Sauvignon) and 30.73 ± 0.64 (Pinot Gris) mg GAE/g d.w. (Figure 4b). Cabernet Sauvignon canes crude extracts (CS-CE) showed a lower non-flavonoid content, of 7.85 mg GAE/g d.w., while the highest content of non-flavonoids was recorded in the canes of the Pinot Gris cultivar (10.44 mg GAE/mg d.w.). Also, the canes of the Pinot Gris cultivar showed the highest content of flavonoid phenolic compounds (20.29 mg GAE/mg d.w.).

Similar concentrations of phenolic compounds in grapevine canes have been reported by other authors. Using an 80% (*v/v*) ethanol solution, Noviello et al. [6] reported a total phenolic content between 14.70 ± 0.20 and 36.90 ± 2.20 mg GAE/g d.w. in the canes of 23 grapevine cultivars from Italy, with no correlation with grape color (white or red) or the direction of production (wine or table grapes). Similar TPC values were reported by Çetin et al. [59], in grapevine canes of ten Turkish cultivars (from 25.36 ± 1.62 to 36.56 ± 2.67 mg GAE/g d.w.), by using a 60% ethanol aqueous solution for the extraction of the phenolic compounds.

Due to the large number of structurally similar compounds in the plant material, analysis of individual polyphenolics is relatively difficult and complicated. Various amounts of interfering material can be extracted together with the phenolic compounds, a purification

step often being required to eliminate the fractions that may interfere with the analysis. A total of 100 mL of each crude extract was concentrated to dryness, redissolved in 0.01% HCl (*v/v*) in distilled water and used for further purification. Phenolic compounds were adsorbed onto the C-18 preconditioned C-18 SPE mini-column, while sugars, acids and other water-soluble compounds were removed by washing the cartridge with 0.01% HCl (*v/v*) in distilled water. The purified extracts (PE) were obtained by the elution of the cartridge sorbent with 96% ethanol. Reported against the initial solid plant material (dry weight), it was observed that the total concentrations of phenolic compounds were lower after the purification procedure. The highest differences were found in the Pinot Gris cultivar canes (−6%), the class of non-flavonoid compounds being more affected (Figure 4b). In the case of purified extracts, the total flavonoid content varied between 17.39 (Cabernet Sauvignon) and 20.40 (Pinot Gris) mg GAE/g d.w., while the non-flavonoids showed concentration from 6.75 to 8.45 mg GAE/g d.w.

The presence of proteins in the crude extracts was highlighted by the Biuret reaction. A light purplish-violet coloration was produced for all samples, indicating the complex formation between the cupric ions and the peptide bond. According to Shen [75] the Biuret reaction is independent of the protein composition, but protein association state may influence the results obtained with the Biuret reagent. The purification step led to the elimination of protein-type interferences; a fact highlighted by the lack of coloration in the purified extracts (image not shown).

### 3.4. Chlorophyll and Carotenoid Content

Most woody plant parts possess, under the peridermic or rhytidome outer layers, pale-green tissues (chlorenchyma) containing assimilatory pigments [76]. Chlorophyll and carotenoid pigments are essential compounds in light energy conversion, being found in the highest amounts in leaves and green shoots. In woody canes the photosynthetic pigments (chlorophyll a and b and total carotenoids) were found in very low concentrations. The absorbance in the visible spectrum (400–800 nm) of the crude ethanolic extracts is shown in Figure 5, the specific wavelengths for the analyzed compounds being marked.

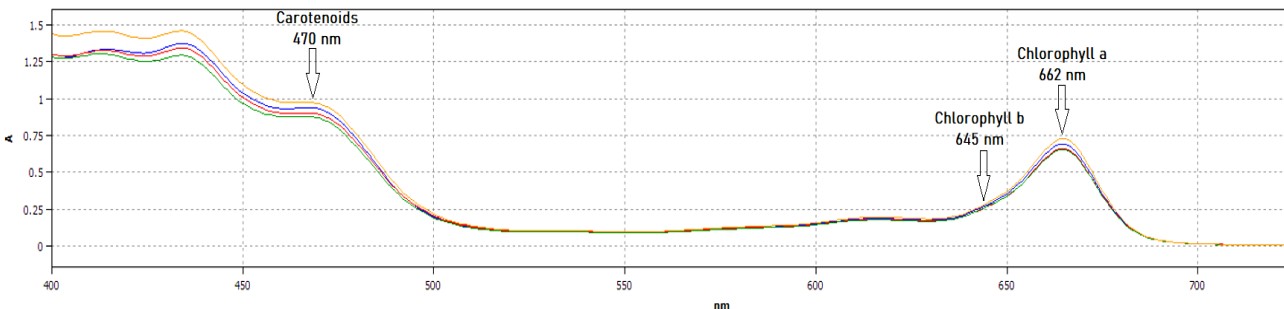

**Figure 5.** Absorption spectra of *Vitis vinifera* L. cultivars cane ethanolic (96% *v/v*) extracts. Blue line—Sauvignon Blanc; red line—Cabernet Sauvignon; green line—Pinot Gris; yellow line—a mixture of plant extracts (as control).

Chlorophyll a (Chl a) was determined in the highest concentrations, between 53.84 ± 0.42 (Pinot Gris) and 58.03 ± 0.57 (Sauvignon Blanc) μg/g d.w., followed by total carotenoid content (Table 2).

The concentrations of assimilatory pigments, the chlorophyll a/chlorophyll b ratio and chlorophyll/carotenoids ratio varied significantly depending on the cultivar. It is known that, in grapevine leaves, the variation in chlorophyll a/b and chlorophyll/carotenoids ratios is an indicator of senescence, stress and damage to the photosynthetic apparatus [41], but little is known about the role and implication of the residual amounts of photosynthetic pigment in the woody grapevine canes, further studies being necessary.

**Table 2.** Chlorophyll and carotenoid content of dry grapevine canes.

| Parameter | Grapevine cultivar | | |
| --- | --- | --- | --- |
| | **Sauvignon Blanc** | **Pinot Gris** | **Cabernet Sauvignon** |
| Chl a (µg/g d.w.) | 58.03 ± 0.57 a | 53.84 ± 0.42 c | 56.41 ± 0.38 b |
| Chl b (µg/g d.w.) | 16.12 ± 0.19 c | 16.79 ± 0.14 b | 17.15 ± 0.22 a |
| Car (x + c) (µg/g d.w.) | 27.71 ± 0.21 a | 22.97 ± 0.26 c | 24.63 ± 0.41 b |
| Chl a/b | 3.60 ± 0.04 a | 3.21 ± 0.02 c | 3.29 ± 0.02 b |
| Chl/Car | 2.68 ± 0.03 c | 3.08 ± 0.01 a | 2.99 ± 0.03 b |

Note: Chl—chlorophyll; Car (x + c)—carotenoids (xanthophylls and carotenes). Mean values with standard deviation (±). Different letters indicate significant differences in Duncan's multiple range test.

### 3.5. HPLC-PDA Polyphenolic Profile

The chromatographic analysis of the purified polyphenolic extracts of grapevine canes highlighted the presence in higher concentrations of seven compounds, belonging to the classes of phenolic acids (4-hydroxybenzoic acid, coumaric acid and sinapic acid), flavan-3-ols (catechin and epicatechin), stilbenes (resveratrol) and flavonols (quercetin) (Figure 6).

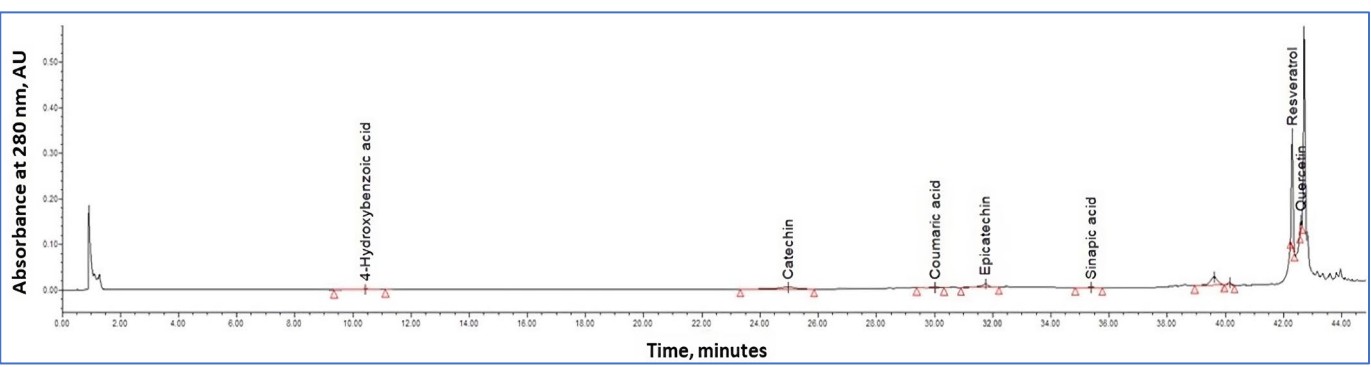

**Figure 6.** HPLC-PAD profile of the grapevine (*V. vinifera* L.) canes purified polyphenolic extracts.

Grapevine canes contained high amounts of resveratrol, the concentrations varying significantly between red-black grape cultivars and white grape cultivars. The highest amount of resveratrol was found in the canes of Cabernet Sauvignon (425.60 µg/g d.w.), closely followed by Pinot Gris cultivar (419.01 µg/g d.w.) (Table 3). Catechin and epicatechin are isomeric flavonoids with proven anti-cancer, anti-inflammatory and anti-viral properties, mainly related to their high number of hydroxyl groups [77]. After resveratrol, catechin (273.50–334.58 µg/g d.w.) and epicatechin (169.91–271.41 µg/g d.w.) were the most abundant phenolic compounds in the *V. vinifera* L. canes. Catechin was found in higher quantity in the canes of Pinot Gris, while epicatechin was in higher concentrations in the canes of the Sauvignon Blanc cultivar. In the case of the flavonoid quercetin, the highest amounts were determined in the dry canes of the Sauvignon Blanc and Cabernet Sauvignon cultivars (72–77 µg/g d.w.).

Phenolic acids were found in the lowest concentrations (<12 µg/g d.w.). Sinapic acid, a hydroxycinnamic acid derivative, was detected only in the extracts of the Sauvignon Blanc cultivar (11.98 µg/g d.w.). Summing the individual amounts of polyphenols, a significantly higher value was registered in the case of Pinot Gris cv. (1001.93 µg/g d.w.).

In the last years, phenolic compounds from grapevine canes were intensively studied, the phenolic profile composition and the concentrations of individual compounds varying significantly depending on the genetic factor (species and cultivar), the growing area, harvest year or the extraction method [16,29,78–80]. Lower concentrations of resveratrol were reported by Nèmeth et al. [79] in canes of Cabernet Sauvignon (6.70–39.10 µg/g f.w.) and Italian Riesling (11.60–207.9 µg/g f.w.) cultivars growing in Hungary. Much higher

levels of resveratrol (up to 4.99 mg/g) were reported by Tříska et al. [81] in the canes of *Vitis vinifera* L. cultivars from the Czech Republic, while Lambert et al. [78] also found higher concentrations of resveratrol in the canes of Pinot noir (1526 ± 293 μg/g d.w.) and Merlot (1181 ± 189 μg/g d.w.) cultivars growing in France, and significantly lower concentration in the canes of the Chardonnay cultivar (190 ± 87 μg/g d.w.), with an average level for all the cultivars of 791 μg/g d.w.

**Table 3.** Individual concentration of the most abundant phenolic compounds identified in grapevine canes' purified extracts.

| Phenolic compounds (μg/g d.w.) | Cultivar | | |
|---|---|---|---|
| | Sauvignon Blanc | Pinot Gris | Cabernet Sauvignon |
| 4-hydroxybenzoic acid | 1.50 ± 0.09 b | 5.68 ± 0.41 a | 6.34 ± 0.30 a |
| Catechin | 321.11 ± 2.07 b | 334.58 ± 2.72 a | 273.50 ± 2.17 c |
| Coumaric acid | 2.69 ± 0.32 b | 10.18 ± 1.54 a | 9.13 ± 0.64 a |
| Epicatechin | 271.41 ± 4.21 a | 177.99 ± 4.08 b | 169.91 ± 4.25 b |
| Sinapic acid | 11.68 ± 1.25 | n.d. | n.d. |
| Resveratrol | 282.19 ± 4.14 b | 419.01 ± 4.10 a | 425.60 ± 5.98 a |
| Quercetin | 77.99 ± 2.50 a | 54.49 ± 4.02 b | 72.75 ± 2.96 a |
| Sum of individual polyphenols | 968.57 ± 14.58 b | 1001.93 ± 19.93 a | 957.23 ± 16.87 b |

Note: Values are presented as the mean of three determinations (n = 3), with standard deviations (±). n.d.—not detected. Different letters indicate significant differences in Duncan's multiple range test ($p \leq 0.05$).

When comparing the values, it should also be considered that high concentrations of resveratrol can also appear as a response of plants to diseases or injury stresses [40], stilbenes acting as phytoalexins protecting plants against aggression of pathogenic microorganisms [82].

The presence of phenolic acids, including 4-hydroxybenzoic acid, coumaric acid and sinapic acid in the phenolic profile of the grapevine canes was also reported by several authors [18,29,40,59,83–85], their concentration varying widely depending on the cultivar and the extraction method. Slightly higher concentrations of catechin and epicatechin were usually reported in grapevine canes, ranging between 204 and 326 μg/g d.w. for epicatechin and between 1150 and 2890 μg/g d.w. for catechin in the canes of some grapevine cultivars from Portugal, extracted using subcritical water [84], and up to 4900 μg/g d.w. in the acetonitrile fraction of *V. vinifera* L. cultivar canes from Italy [86].

### 3.6. Antioxidant and Antiradical Potential

Although other waste from the winemaking industry (pomace, seeds, stems, etc.) have often been tested for their antioxidant and antiradical activities, very few studies have been carried out to evaluate the biologically active properties of grapevine canes. Previous research concluded that extractions with aqueous alcoholic mixtures contribute to the increase in antioxidant capacity, the presence of water facilitating the release of some hydrophilic antioxidants [67,87]. In traditional medicine whole plants or mixtures of plants are usually used, with crude extracts often showing a higher biological activity than isolated constituents at an equivalent dose [88]. Thus, both crude and purified extracts of grapevine canes were tested for their antioxidant and antiradical activities. In order to compare the results with those obtained for the standard antioxidant compounds (gallic acid and ascorbic acid), all samples were prepared and tested in similar concentrations (0–100 ug/mL). According to Burlakova et al. [89], the antiradical activity characterizes the ability of compounds to react with free radicals, while antioxidant activity represents the ability to inhibit the process of oxidation. Also, reductive capabilities of plant extracts can serve as an indicator of their potential antioxidant activity [38].

### 3.6.1. DPPH Scavenging Activity

The DPPH (2,2-diphenyl-1-picryl-hydrazyl) assay is one of the most common procedures used to evaluate the free radical scavenging activity of various compounds or plant extracts, due to its practical significance, affordability, simplicity and reproducibility [90]. In the DPPH radical-scavenging assay, antioxidants react with DPPH, and convert it to the yellow-colored diphenyl picrylhydrazine. The color fading and decrease in absorbance at 517 nm are directly correlated with the scavenging power of the tested extracts [91]. The DPPH scavenging effect of the grapevine cane extracts and standard antioxidants were plotted on a graph in relation to the cultivar and extract concentration (Figure 7a). Grapevine cane crude and purified extracts showed a dose-dependent antioxidant activity against DPPH (%) ($r^2$ = 0.9640–0.9897). However, the crude extracts of all three cultivars, regardless of the concentration, presented a lower DPPH scavenging activity compared to the purified polyphenolic extracts. Thus, at a concentration of 60 µg/mL, the values were up to 60% higher for the purified extracts. Also, for all tested concentrations, the DPPH scavenging capacity of the crude extracts showed a non-significant variation between cultivars (<20%). However, the highest antioxidant capacity was shown by the crude and purified extracts of the Pinot Gris cultivar.

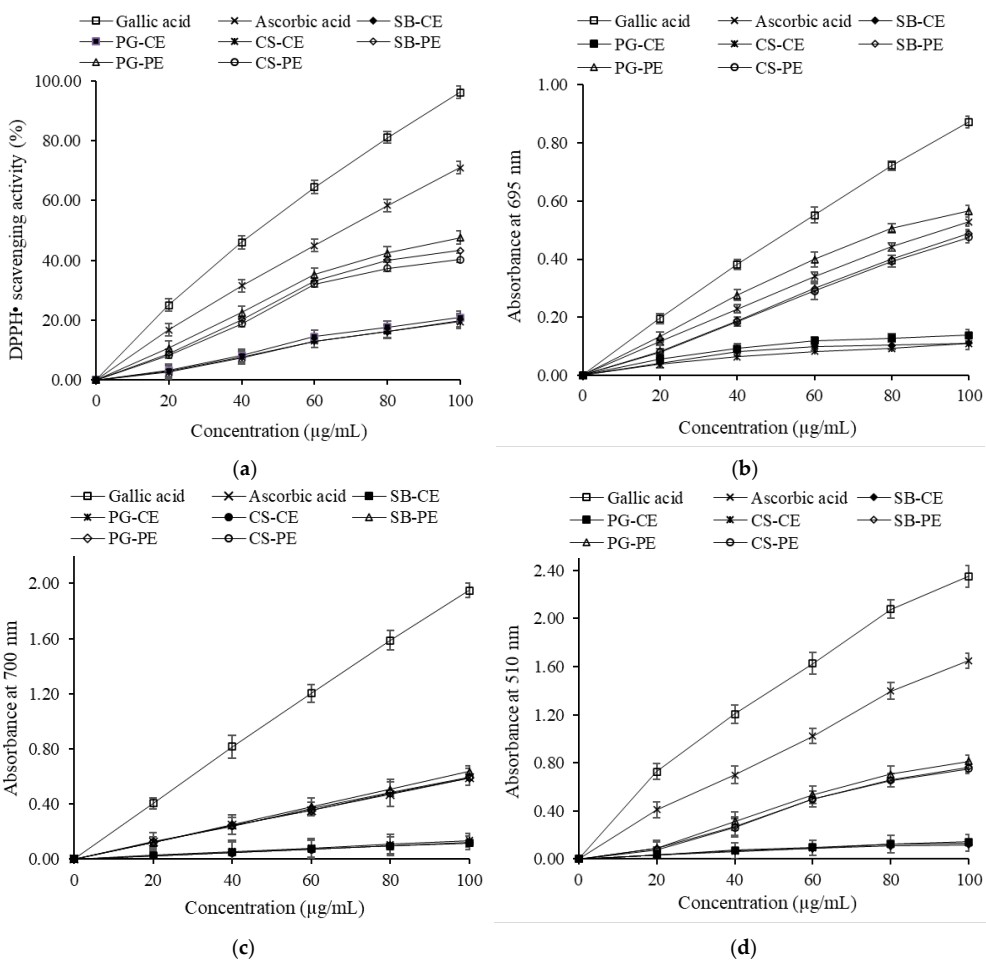

**Figure 7.** DPPH scavenging activity (**a**), phosphomolybdate scavenging capacity (**b**), ferric reducing power (**c**) and phenanthroline reduction (**d**) of crude and purified ethanolic extracts of *V. vinifera* L. dried canes and positive controls (gallic acid and ascorbic acid) (0–100 µg/mL). Note: DPPH—2,2-diphenyl-1-picryl-hydrazyl free radical; SB-CE—Sauvignon Blanc-crude extract; PG-CE—Pinot Gris-crude extract; CS-CE—Cabernet Sauvignon-crude extract; SB-PE—Sauvignon Blanc-purified extract; PG-PE—Pinot Gris-purified extract; CS-PE—Cabernet Sauvignon-crude extract. Values are presented as mean of three determinations (n = 3), with standard deviations (error bars).

The antioxidant activity of the highest concentration tested (100 μg/mL) was also expressed as ascorbic acid equivalents (AAE) by interpolation from the linear regression analysis, in order to evaluate the relationship between the scavenging power of the cane extracts and standard antioxidants. Thus, at 100 μg/mL, the purified polyphenolic extracts of the Pinot Gris cultivar showed the highest DPPH scavenging activity (47.60 ± 2.15% scavenged DPPH, respectively, 64.92 ± 1.36 μg AAE/mL) (Figures 7a and 8a).

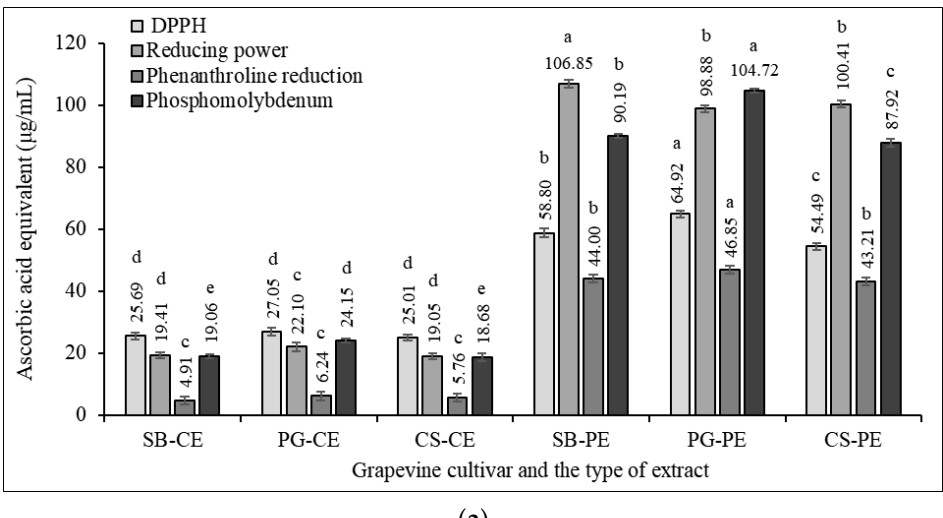

(**a**)

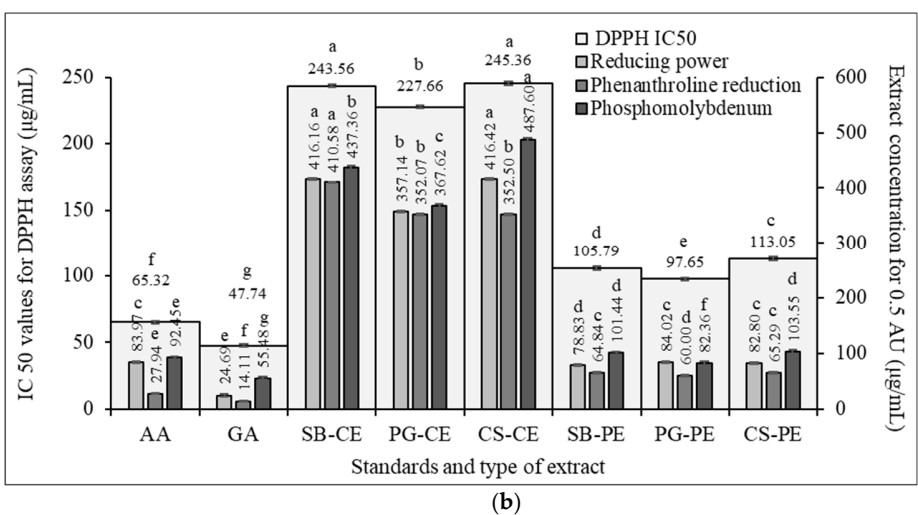

(**b**)

**Figure 8.** Ascorbic acid equivalent (AAE) of antioxidant/antiradical activity of the extracts at the highest tested concentration (100 μg/mL) (**a**); and IC 50 values (for DPPH scavenging activity) and effective concentration at which the absorbance was 0.5 (secondary axis; for ferric reducing power, phenantroline and phosphomolybdenum assays) (**b**). Note: DPPH—2,2-diphenyl-1-picryl-hydrazyl free radical; SB-CE—Sauvignon Blanc-crude extract; PG-CE—Pinot Gris-crude extract; CS-CE—Cabernet Sauvignon-crude extract; SB-PE—Sauvignon Blanc-purified extract; PG-PE—Pinot Gris-purified extract; CS-PE—Cabernet Sauvignon-crude extract. Values are presented as the mean of three experiments (n = 3), with standard deviations (error bars). Different letters indicate significant differences in Duncan's multiple range test ($p \leq 0.05$).

The IC50s, as the concentration of extract required to scavenge 50% of the initial DPPH free radical, showed significantly lower values for the purified cane ethanolic extracts (97.65–113.05 μg/mL) (Figure 8b). The lower the IC50 value, the more potent was the substance in scavenging DPPH, indicating a higher antioxidant activity [92]. The IC50 values varied significantly between the cultivars and type of extract, but also compared to the standard antioxidant compounds, gallic acid showing the lowest values. Zhang et al. [38]

reported IC50 values between 44.00 and 60.88 µg/mL for the cane ethanolic extracts of three French *Vitis vinifera* L. wine cultivars, directly correlated with the phenolic content of the samples.

### 3.6.2. Phosphomolybdenum Assay

The phosphomolybdenum assay was used to evaluate the total antioxidant capacity of the grapevine cane extracts. The basic principle involves the reduction of Mo (VI) to Mo (V) by the tested plant extracts. The total antioxidant activity was expressed as the absorbance of the sample at 695 nm, a higher absorbance value indicating higher antioxidant activity [93]. The cane extracts showed a total antioxidant capacity that gradually increased with the increase in extract concentration (Figure 7b). The purified extracts showed a higher antioxidant activity compared to the crude extracts, by up to 77%. However, gallic acid showed the highest antioxidant activity in the phosphomolybdenum assay. It should be noted that the purified extract of Pinot Gris canes presented a higher activity compared to ascorbic acid, even at very low concentrations. At 100 µg/mL, the phosphomolybdenum assay indicated significant differences between the crude (0.109–0.138) and purified (0.476–0.565) cane extracts.

Equating in ascorbic acid equivalents (AAE), the purified extract of Pinot Gris canes showed the highest value among the tested extracts (104.72 ± 0.73 µg/mL), followed by the purified extracts of Sauvignon Blanc (90.19 ± 0.61 µg/mL) and Cabernet Sauvignon (87.92 ± 1.33 µg/mL) (Figure 8a). The IC50 values (µg extract/mL) as the effective concentration at which the absorbance at 695 nm reached 0.5, was calculated by interpolation from the linear regression analysis. For the crude extracts, the effective concentration was high, varying significantly between cultivars; the highest antioxidant activity (the lowest IC50 value) being obtained for the Pinot Gris cultivar (PG-CE; 367.62 ± 3.01 µg/mL), which was about four times higher compared to the purified extract of the same cultivar (82.36 ± 3.24 µg/mL) (Figure 8b).

As far as we have found, there are no studies that evaluate the antioxidant activity of grapevine canes by the phosphomolybdenum assay, although several plant sources have been evaluated. Thus, Moonmun et al. [46], reported IC50 values between 125.77 and 178.79 µg/mL for the ethanolic extracts of *Heliconia rostrata* rhizomes, while Kaushik et al. [94] calculated an IC50 of 435.70 µg/mL for a polyherbal methanolic extract.

### 3.6.3. Ferric Reducing Power

In the ferric reducing assay, the compounds which exhibit a reduction potential react with potassium ferricyanide ($Fe^{3+}$) to form potassium ferrocyanide ($Fe^{2+}$), which reacts further with ferric chloride to form a ferric ferrous complex (pale green to blue color) that has an absorption maximum at 700 nm [95]. The increase in the absorbance of the reaction mixture indicates a higher antioxidant activity in the samples. All ethanolic extracts were capable of reducing $Fe^{3+}$ in a linear dose-dependent model (Figure 7c). Gallic acid, as standard, showed the highest ferric reducing power. Also, the purified extracts exhibited a higher reducing activity, which increased with the extract concentration. At 100 µg/mL the purified extracts showed an absorbance that varied between 0.597 ± 0.050 (Cabernet Sauvignon) and 0.635 ± 0.065 (Sauvignon Blanc), similar to that of ascorbic acid (0.589 ± 0.041). Half maximal effective concentrations, such as the effective concentration at which the absorbance was 0.5, were significantly higher for the crude extracts (357.14–416.42 µg/mL), while for the purified extracts the values did not exceed 85 µg/mL, showing a stronger ferric reducing ability (Figure 8b).

### 3.6.4. Phenanthroline Assay

The phenanthroline assay is based on the reduction of the $Fe^{3+}$ ion to the $Fe^{2+}$ ion by an antioxidant, the formed $Fe^{2+}$ ion reacting with ortho-phenanthroline to form a red-orange complex measured at 510 nm [96]. The relationship between the extract concentration and the absorbance of the reaction mixtures is plotted in Figure 7d. Both the crude and purified

extracts showed lower values compared to the standard compounds (gallic and ascorbic acids). Among the tested samples, the highest absorbance was recorded in the case of the purified extracts of the Pinot Gris canes. At the highest extract concentration tested (100 μg/mL), the equivalence in ascorbic acid indicated an antioxidant activity that was 78% higher for the purified polyphenolic extracts (43.21–46.25 μg AAE/mL), compared to the crude extracts (4.91–5.76 μg AAE/mL) (Figure 8a). The IC50, calculated as the effective concentration at which the absorbance was 0.5, showed a significantly lower value in the case of the Pinot Gris purified extract, 60.00 ± 1.74 μg/mL, while for ascorbic acid the absorbance of 0.5 was reached at a concentration of 27.94 ± 1.21 μg/mL and for gallic acid at 14.11 ± 0.94 μg/mL (Figure 8b).

3.6.5. Antioxidant/Antiradical Activity and Phenolic Compounds' Correlation

Lower IC50 values indicate higher radical-scavenging power of the extracts. As shown in Figure 8b, the rank order of IC50 values for the DPPH assays was: gallic acid (GA) > ascorbic acid (AA) > Pinot Gris purified extract (PG-PE) > Sauvignon Blanc purified extract (SB-PE) > Cabernet Sauvignon purified extract (CS-PE) > Pinot Gris crude extract (PG-CE) > Sauvignon Blanc crude extract (SB-CE) > Cabernet Sauvignon crude extract (CS-CE). For the phosphomolybdenum antioxidant assay the order was modified as follows: GA > PG-PE > AA > SB-PE > CS-PE > PG-CE > CS-CE > SB-CE. Ferric reducing power highlighted again the antiradical power of the purified extracts: GA > SB-PE > CS-PE > AA > PG-PE > PG-CE > SB-CE > CS-CE. The effective concentration at which the absorbance was 0.5 in the phenanthroline assay indicated a different order in the extract antiradical capacity, respectively: GA > AA > SB-PE > CS-PE > PG-PE > PG-CE > SB-CE > CS-CE.

Data correlation (Pearson correlation coefficient r) showed that the IC50 values of the grapevine cane extracts were inversely correlated with their total phenolic, flavonoid and non-flavonoid contents, both in the case of crude and purified extracts (Table 4). Normally, a higher content of phenolic compounds in the extract corresponded to lower IC50 values. Thus, TPC was inversely correlated with IC50 values of DPPH scavenging activity of crude (r = −0.9689) and purified extracts (r = −0.9831), with the efficient concentration of purified extracts in the ferric reducing power assay (r = −0.9689), with the efficient concentration of crude extracts (r = −0.9866) in the phenanthroline reduction assay, and with the efficient concentration of crude (r = −0.9887) and purified (r = −0.9883) extracts in the phosphomolybdenum assay, supporting the former statement on the contribution of phenolic compounds in the antioxidant and antiradical activities of grape cane extracts. A similar trend was registered in the case of the correlation of the two main groups of phenolic compounds (flavonoids and non-flavonoids) with the IC50 values of antioxidant/antiradical activity (Table 4).

**Table 4.** Correlation coefficients between the antioxidant/antiradical capacity (as IC50) of the grapevine canes' crude and purified extracts and their phenolic content.

| Phenolic Group | DPPH | | FRP | | PR | | PA | |
|---|---|---|---|---|---|---|---|---|
| | CE | PE | CE | PE | CE | PE | CE | PE |
| TPC | −0.9689 | −0.9831 | 0.4877 | −0.9630 | −0.9866 | −0.2506 | −0.9887 | −0.9883 |
| Fl. | −0.9670 | −0.9862 | 0.4946 | −0.9676 | −0.9878 | −0.2676 | −0.9899 | −0.9854 |
| Nfl. | −0.9724 | −0.9794 | 0.4754 | −0.9577 | −0.9842 | −0.2322 | −0.9865 | −0.9910 |
| ∑HPLC | - | −0.9703 | - | 0.4827 | - | −0.9856 | - | −0.9878 |

Note: CE—crude extracts; PE—purified extracts; DPPH—2,2-diphenyl-1-picryl-hydrazyl free radical scavenging activity; FRP—Ferric reducing power; PR—phenanthroline reduction assay; PA—phosphomolybdenum assay; TPC—total phenolic compounds; Fl.—flavonoids; Nfl.—non-flavonoids; ∑$_{HPLC}$—sum of individual polyphenols determined by HPLC analysis.

Regarding the correlation of the sum of individual phenolic compounds determined by HLPC with the values of the half maximal inhibitory concentration, it was observed

that high phenolic contents were correlated with lower IC50 values (Table 4). According to the correlations, catechin contributed most to the antioxidant activity of the extracts (DPPH; r = −0.9410), while high concentrations of epicatechin, quercetin and sinapic acid were correlated with a high antiradical activity (FRP; r = −0.8218 to −0.9558). The results indicate that phenolic compounds of grapevine canes are important constituents that have the ability to scavenge the free radicals. These results are consistent with those reported by Zhang et al. [38] and Ju et al. [16] for various grapevine cultivar cane extracts, highlighting the relationships between the phenolic content and the antioxidant/antiradical activities.

### 3.7. Antibacterial Susceptibility Testing

Resistance to antimicrobial products is a global health crisis and a serious threat to the health of the population, considering that there are bacterial strains that have acquired resistance to almost all antibiotics. Therefore, the identification of new antibacterial agents is a continuous process that is necessary to overcome resistant bacteria [97]. Disk diffusion susceptibility testing is a standardized technique widely used for antibacterial compounds in routine clinical microbiology laboratories [98]. If tested plant extracts are microbiologically active, an inhibition zone develops around the filter paper disk after incubation, whose diameter indicates the antimicrobial potency of the extracts [99]. In our studies, bacterial strains of *E. coli* (G−) and *S. aureus* (G+) were used to test the antimicrobial activity of grapevine cane crude and purified extracts (10–100 mg/mL). Cultures were grown aerobically for 24 h at 37 °C, and the antibacterial capacity was calculated as the mean zone of inhibition (mm). DMSO (5%) and gentamicin (10 μg/mL) were negative and positive controls, respectively.

The diameters of the inhibition zones were small but increased along with the extract concentration. The best results were obtained in the case of purified extracts, for both *S. aureus* (Figure 9a) and *E. coli* (Figure 9b). According to Koh et al. [100] a purified extract with improved bioactivity is highly desirable, to reach an effective dose range in a practical dosage, the purification often being associated with the removal of highly extractable but biologically inactive polysaccharides.

For *S. aureus*, at the highest concentration of extract (100 mg/mL), the diameter of the inhibition zone varied significantly between the crude (2.20–2.40 mm) and purified (3.40–4.10 mm) polyphenolic extracts. In the case of the Gram-negative bacteria (*E. coli*), although at 100 mg/mL the crude extracts showed a slightly higher activity (2.40–2.50 mm) compared to *S. aureus*, the purified polyphenol extracts induced smaller zones of inhibition (3.20–3.80 mm) than in the case of the Gram-positive bacteria.

The better action of the polyphenolic extracts against the Gram-positive bacteria is not surprising, considering their distinctive structure. Due to its peptidoglycan cell wall, surrounded by an outer lipopolysaccharide membrane, Gram-negative bacteria are more resistant to antibacterial compounds than Gram-positive bacteria [97,101].

The purified extract of the Pinot Gris canes (PG-PE) induced the largest zone of inhibition for both *S. aureus* (Figure 9a) and *E. coli* (Figure 9b), with higher values in the case of the Gram-positive bacteria (>4 mm). A similar situation was also observed in the case of the purified extracts from the canes of Sauvignon Blanc (SB-PE) and Cabernet Sauvignon (CS-PE) cultivars, however, the values were significantly lower compared to purified extract of Pinot Gris (PG-PE) canes. The inhibition zones obtained for different concentrations of purified extract can be observed in Figure 10a,b.

MIC values were high, varying within small limits between cultivars. *S. aureus* proved to be more sensitive to the action of the cane extracts compared to the Gram-negative bacteria *E. coli.* Both in the case of *E. coli* (70 mg/mL) and *S. aureus* (60 mg/mL) the purified polyphenolic extracts showed lower MIC values (Table 5).

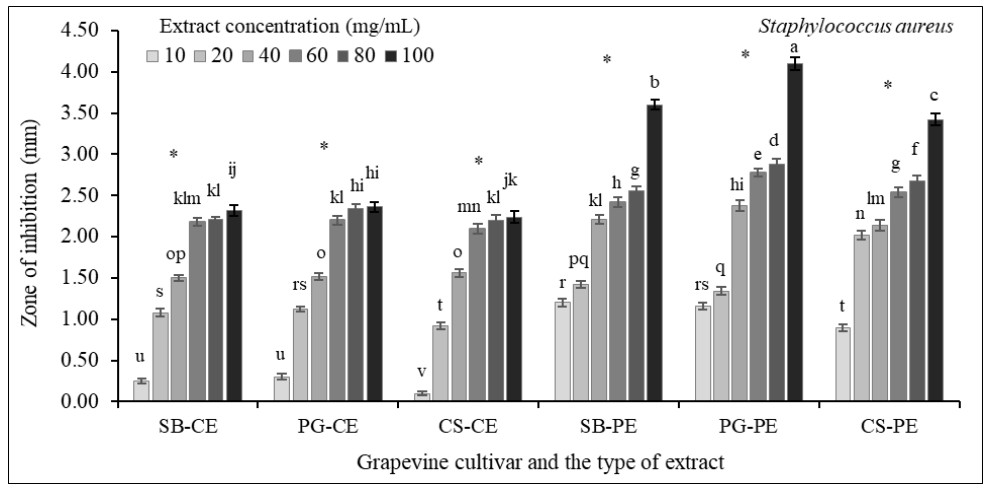

(**a**)

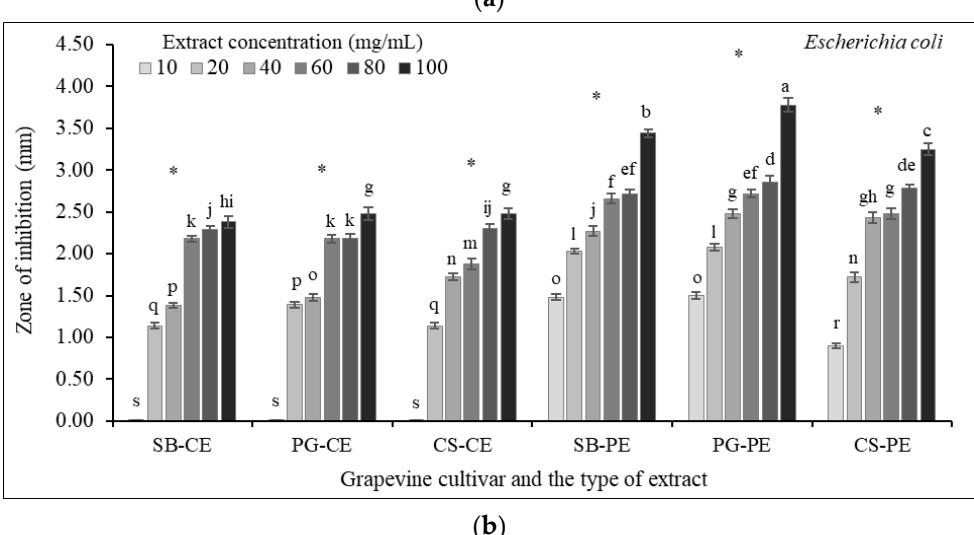

(**b**)

**Figure 9.** Antimicrobial activity of grapevine cane crude and purified extracts assessed by measuring the diameter zones of inhibition in mm (disk-diffusion method): (**a**) *S. aureus* and (**b**) *E. coli*. Note: SB-CE—Sauvignon Blanc-crude extract; PG-CE—Pinot Gris-crude extract; CS-CE—Cabernet Sauvignon-crude extract; SB-PE—Sauvignon Blanc-purified extract; PG-PE—Pinot Gris-purified extract; CS-PE—Cabernet Sauvignon-crude extract. * indicates significant differences inside the data set (ANOVA; $p < 0.05$), while different letters indicate significant differences between the mean values (Duncan's multiple range test). Error bars represent the standard deviation between three independent experiments data (n = 3).

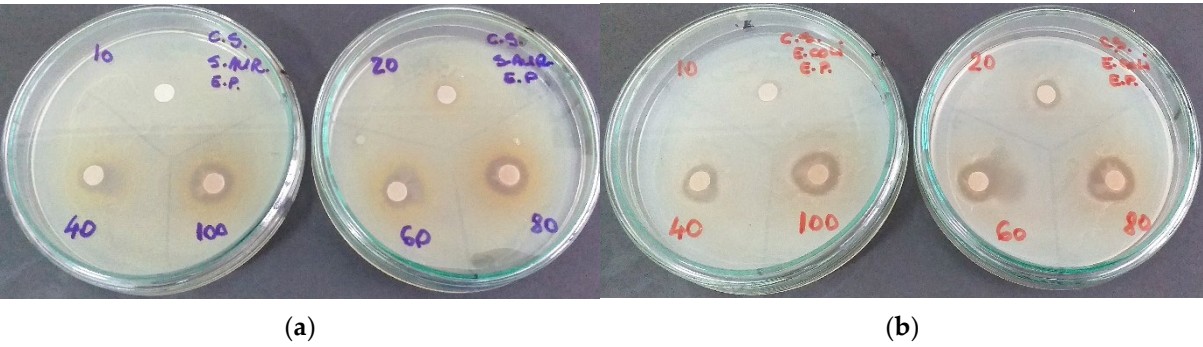

(**a**)                        (**b**)

**Figure 10.** Antimicrobial activity of increasing concentrations of purified grapevine cane polyphenolic extracts (10–100 mg/mL) on *S. aureus* (**a**) and *E. coli* (**b**).

**Table 5.** MIC, MBC and MIC index of grapevine cane crude and purified extracts on *S. aureus* and *E. coli* bacteria.

| Sample | *Staphylococcus aureus* | | | *Escherichia coli* | | |
|---|---|---|---|---|---|---|
| | **MIC (mg/mL)** | **MBC (mg/mL)** | **MIC Index** | **MIC (mg/mL)** | **MBC (mg/mL)** | **MIC Index** |
| SB-CE | 80 ± 0 | 110 ± 0 | 1.38 | 90 ± 0 | 120 ± 0 | 1.33 |
| PG-CE | 80 ± 0 | 100 ± 0 | 1.25 | 90 ± 0 | 110 ± 0 | 1.22 |
| CS-CE | 80 ± 0 | 110 ± 0 | 1.38 | 100 ± 0 | 120 ± 0 | 1.20 |
| SB-PE | 60 ± 0 | 100 ± 0 | 1.67 | 70 ± 0 | 110 ± 0 | 1.57 |
| PG-PE | 60 ± 0 | 90 ± 0 | 1.50 | 70 ± 0 | 100 ± 0 | 1.43 |
| CS-PE | 60 ± 0 | 100 ± 0 | 1.67 | 70 ± 0 | 120 ± 0 | 1.71 |

Note: MIC—Minimum inhibitory concentration; MBC—minimum bactericidal concentration; MIC index— MBC/MIC ratio; SB-CE—Sauvignon Blanc—crude extract; PG-CE—Pinot Gris—crude extract; CS-CE—Cabernet Sauvignon—crude extract; SB-PE—Sauvignon Blanc-purified extract; PG-PE—Pinot Gris—purified extract; CS-PE—Cabernet Sauvignon-crude extract. Mean values of three independent experiments (n = 3), with standard deviation (±).

MBC, as the lowest extract concentration that prevents the growth of bacteria, varied within much wider limits depending on the type of extract. In the case of the crude extracts (CE) both microorganisms were inhibited at high concentration between 100 and 120 mg/mL. The use of purified extracts led to a lowering in MBC, the values varying between 90 and 100 mg/mL for *S. aureus* and up to 120 mg/mL for the *E. coli* strain.

Antibacterial susceptibility was also represented according to the MBC/MIC ratio as the MIC index. If the ratio MBC/MIC ≤ 4, the effect is considered bactericidal, while at a ratio MBC/MIC > 4, the effect is considered bacteriostatic [56]. Taking into account the values of this ratio, all extracts showed a bactericidal effect on the tested strains (MIC index < 4) (Table 5).

Several authors demonstrated the antimicrobial activity of some plant extracts on the two bacterial species, MIC values varying, depending on the polyphenol composition, from 7–50 mg/mL [102–104], to values over 300 mg/mL [105]. Despite the increased interest in recent years in the utilization of natural products with antimicrobial effects, information concerning the antimicrobial activity of grapevine cane extracts is still scarce. Gullón et al. [106] evaluated the antimicrobial activity of extracts from the grapevine shoot liquors against several microorganisms associated with spoilage of food products, reporting MIC values ranging from 5 to 20 mg/mL. Also, Moreira et al. [40] demonstrated that grapevine shoot extracts (microwave-assisted extraction) of two *V. vinifera* L. cultivars from Portugal possess antimicrobial activity against bacteria (*E. coli* and *Streptococcus mitis*) and yeasts (*Candida albicans*), the MIC values being <20 mg/mL.

In the case of microorganisms of oenological interest (yeasts and lactic acid bacteria), the effect of the grapevine cane extracts (in 5% DMSO) on their development was very low (<3 mm diameter of the inhibition zone) (Figure 11a,b). However, the antimicrobial activity against lactic acid bacteria was more effective than against yeast. Unlike pathogenic bacteria, both yeast (*Saccharomyces cerevisiae*) and lactic acid bacteria (*Oenococcus oeni*) growth were inhibited rather by the high concentrations of crude extracts (1.00–1.50 mm for *S. cerevisiae* and 1.80–2.20 mm for *O. oeni*) (Figure 11a), than by the purified polyphenolic extracts (0.60–1.00 mm for *S. cerevisiae* and 0.90–1.10 mm for *O. oeni*) (Figure 11a). However, for both yeast and lactic acid bacteria strains, the MIC values were over 120 mg/mL.

Grapevine canes and implicitly, the crude ethanolic extracts, were shown to be rich in various secondary metabolites [59,107,108]. Even if traditional medicine often uses the extracts or decoctions from the whole plant or parts of it, the selective extraction and use of compounds with tested and proven biological value seems preferable. However, further studies are necessary to elucidate the mechanisms that can underlie the synergistic effect of grapevine cane polyphenols and different classes of compounds on the growth and development of microorganisms used in winemaking bioprocesses. Also, additional studies

are needed to highlight the biological activity of grapevine cane extracts on various strains of pathogenic microorganisms, in the current context of increasing antibiotic resistance.

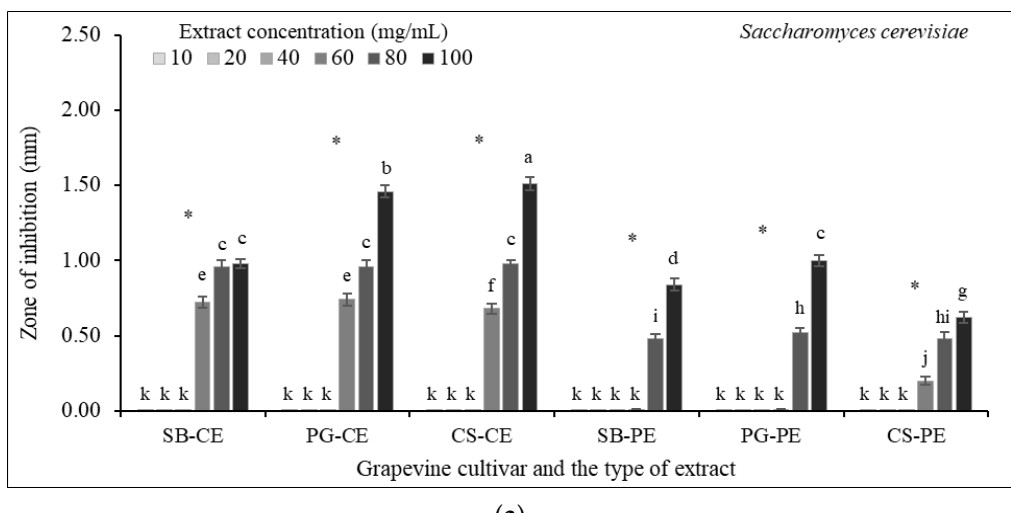

(**a**)

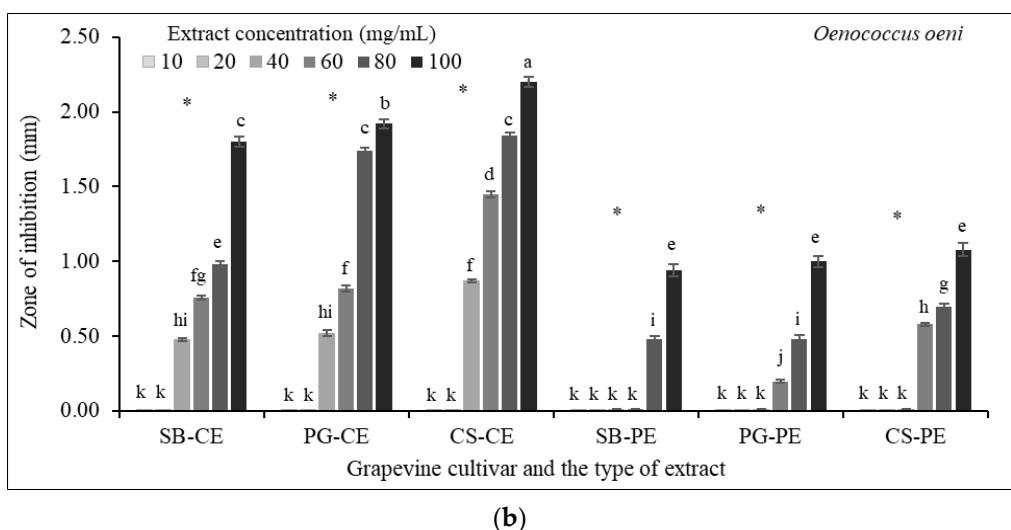

(**b**)

**Figure 11.** Antimicrobial activity of *V. vinifera* L. cane crude and purified extracts against *Saccharomyces cerevisiae* (**a**) and *Oenococcus oeni* (**b**), assessed by measuring the diameter zones of inhibition in mm (disk-diffusion method). Note: SB-CE—Sauvignon Blanc-crude extract; PG-CE—Pinot Gris-crude extract; CS-CE—Cabernet Sauvignon-crude extract; SB-PE—Sauvignon Blanc-purified extract; PG-PE—Pinot Gris-purified extract; CS-PE—Cabernet Sauvignon-crude extract. * indicates significant differences inside the data set (ANOVA test; $p < 0.05$), while different letters indicate significant differences between the mean values (Duncan's multiple range test). Error bars indicate the standard deviation between three independent experiments data (n = 3).

## 4. Conclusions

Grapevine canes harvested during spring pruning proved to be a valuable source of carbohydrates, minerals and more importantly, phenolic compounds. The highest yield of phenolic compounds which maintained their antioxidant activity was obtained when the ground dry canes (<0.5 mm) were subjected to liquid–solid extraction with 70% (*v*/*v*) ethanol solution, in stationary, for 4 h at 35 °C, after a preliminary ultrasound treatment (6 min., 42 KHz). The highest amounts of phenolic compounds, both flavonoids and non-flavonoids, were determined to be in the canes of the Pinot Gris cultivar, the first extraction fraction including about 70% of the total phenolic compounds extracted. Chlorophyll and carotenoid pigments were also found, their concentrations and ratios varying significantly between cultivars. HPLC analysis of the extracts indicates that stilbenes

(resveratrol), flavan-3-ols (catechin and epicatechin), flavonols (quercetin) and phenolic acids (4-hydroxybenzoic acid, coumaric acid, sinapic acid) were the main phenolic compounds in the *V. vinifera* L. cultivar canes. Grapevine canes contained high concentrations of resveratrol, the values varying significantly between red-black grape cultivars and the white grape cultivar. The highest amount of resveratrol was found in the canes of Cabernet Sauvignon and Pinot Gris cultivars. The antioxidant and antiradical activities of the purified polyphenolic extracts (C-18 SPE cartridge) were higher compared to crude extracts and showed a higher antimicrobial effect on Gram-negative (*Escherichia coli*) and Gram-positive (*Staphylococcus aureus*) pathogenic bacteria. However, *S. aureus* proved to be more sensitive to the action of the cane extracts compared to the Gram-negative bacterium *E. coli*. On the contrary, yeast (*Saccharomyces cerevisiae*) and lactic acid bacteria (*Oenococcus oeni*) showed reduced susceptibility to the crude and purified polyphenolic extracts even at high concentrations. These findings indicate that grapevine canes represent an accessible and sustainable source of natural bioactive compounds for the food, cosmetic, and pharmaceutical industries, that are currently insufficiently known and unexploited to their true functional and economic potential.

**Author Contributions:** Conceptualization, R.V.F., S.I.D. and A.P.; methodology, R.V.F., A.P., L.C. and F.D.B.; software, L.C. and R.M.F.; validation, S.I.D. and C.I.B.; formal analysis, R.V.F., S.R. and R.M.F.; investigation, R.V.F. and C.I.B.; resources, R.V.F. and A.P.; data curation, R.V.F., S.I.D. and A.P.; writing—original draft preparation, R.V.F. and C.I.B.; writing—review and editing, C.I.B. and S.I.D.; visualization, A.P. and C.I.B.; supervision, R.V.F., S.I.D. and C.I.B.; funding acquisition, C.I.B., F.D.B. and S.R. All authors have read and agreed to the published version of the manuscript.

**Funding:** This research was funded by Research and consulting project no. 5934/17.03.2022.

**Data Availability Statement:** All data is contained within the article.

**Acknowledgments:** The authors gratefully acknowledge the University of Agricultural Sciences and Veterinary Medicine Cluj-Napoca, "Al. I. Cuza" University of Iasi and "Ion Ionescu de la Brad" University of Life Sciences Iasi for the support given in carrying out the research.

**Conflicts of Interest:** The authors declare no conflict of interest.

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
