# Peer review of "Physico-Chemical Characterization, Phenolic Compound Extraction and Biological Activity of Grapevine (Vitis vinifera L.) Canes"

_horticulturae, doi:10.3390/horticulturae9111164_

Round 1

Reviewer 1 Report

Very interesting research on the pharmaceutical value of shoots as a waste product after cutting vines. The publication in this form is too long and should be divided into two parts. The abstract in its current content is unacceptable, it contains general statements, this has been expanded in the summary. The research methodology is very detailed, but I have not learned how samples of shoots were collected after cutting them for laboratory tests, whether from one plant or from many selected from different places in the vineyard, in order to reduce the influence of soil conditions. The results quite often describe again the laboratory procedure that is described in the research methodology, or mark individual treatments with abbreviations, which unnecessarily lengthens the publication. There are no consistency in the spelling of variety names, both parts of the variety name should be in capital letters. The Latin name of the species should be given only at the beginning. I congratulate the authors on the idea and implementation of thorough laboratory research.

English language without any reservations.

Author Response

Reviewer 1

Dear Reviewer,

thank you for the positive comments sent and for the time allocated to review our work.

All changes in the text requested by Reviewer 1 were marked in red.

  1. Abstract - provide percentage changes, as in the results / by what percentage

In the Abstract we highlighted the fact that the differences were statistically significant; the differences in resveratrol concentration are difficult to express as percentage. For completion, the determined concentrations were added in brackets. If numbers are not necessary, we can remove them.

  1. Abstract - .. by what percentage

In this case, is impossible to show the percentage, considering that the antioxidant/antiradical activities were determined by various methods (4); in the Abstract we presented the general trend determined in the study.

  1. Biological material - how shoot samples were taken

Thank you for the observation, more details were added in text.

  1. Biological material - the second part of the variety name written in capital letters / note as above

We changed throughout the text.

  1. Physico-chemical characterization of grapevine canes - use the English name

We modified as requested (“Grapevine” instead of “V. vinifera L.”).

  1. Conclusions - give the Latin name only once at the beginning, why every time?

We have simplified the text by mentioning the species only once.

Reviewer 2 Report

Thanks to the authors for providing this study

This study provides practical information on: “Physico-chemical Characterization, Phenolic Compounds Extraction and Biological Activity of Grapevine (Vitis vinifera L.) canes”.

Main comment:

·       Many experiments and estimates were conducted in this study, but most of the results are known a priori. The amount of content of many components that were estimated was small and useless, and this is expected in a wooden cane. Is there any economic return that can be achieved from the results of this study? That is, is it possible for farmers to use the grape cane resulting from pruning according to the aim of the study and abandon their traditional use of this cane? I think this should be clarified when applying for the purpose of the study.

·       That is, although there are some components in acceptable proportions as mentioned in the results, the cost and method of extracting these materials were not mentioned, so they were not used previously. Because if this was known, it would be possible to compare it with the traditional use of grape cane and achieve greater practical benefit.

However, please answer the following comments:

·       Rephrase the abstract better, so that it begins with an introduction to the topic of study that does not exceed two lines. Then the objective of the study, it cannot be the same as the title of the study as it currently exists. Then the treatments be clearly written. This is followed by a summary of the most important results and finally a recommendation or conclusion. Pay attention to adhering to the number of words specified in the journal’s instructions.

·       Add a paragraph to the introduction about the economic return from using grape cane according to the objective of the study. Citing appropriate references.

·       That was written at the end of the introduction (lines: 149-152) is not appropriate here. It is better to move it to the conclusion where it is more useful.

·       Why were grape cane samples not taken from two seasons?

·       There is something incomprehensible in paragraph 2.5: Has chlorophyll actually been estimated in the pruning residue? Note that it is a one-year-old wooden cane! Where does it get chlorophyll?

·       The results in the paragraph 3.2.5 are not mentioned in the materials and methods (how they were carried out).

·       Separate the results section from the discussion so that the paragraphs discussed are clear.

Moderate editing of English language required

Author Response

Reviewer 2

Respected Reviewer,

thank you very much for your positive comments and the observations sent, that, we are sure, will improve the article.

  1. Rephrase the abstract better, so that it begins with an introduction to the topic of study that does not exceed two lines. Then the objective of the study, it cannot be the same as the title of the study as it currently exists. Then the treatments be clearly written. This is followed by a summary of the most important results and finally a recommendation or conclusion. Pay attention to adhering to the number of words specified in the journal’s instructions.

Changes were made and a general objective was added (sustainable horticulture), considering the limited number of words.

  1. Add a paragraph to the introduction about the economic return from using grape cane according to the objective of the study. Citing appropriate references.

Although the economic evaluation was not the aims of the current study, it was showed in the Introduction that “Considering the high level of waste obtained by pruning in worldwide viticulture, Rayne et al. estimated that the extraction of stilbene from grape canes may reach a global economic value of over $30 billion.” Thank you for this suggestion and we will definitely consider in the next study.

  1. That was written at the end of the introduction (lines: 149-152) is not appropriate here. It is better to move it to the conclusion where it is more useful.

Thank you for this very good observation, but at the end of the Conclusions section it was inserted a similar paragraph. We considered that this phrase at the end of the Introduction is more appropriate as an additional justification of the current study. If necessary, we will move this phrase to another section.

  1. Why were grape cane samples not taken from two seasons?

Initially the purpose of the experiment was to optimize the extraction of phenolic compounds from grapevine canes. Later, at the suggestion of our collaborators (biochemists and microbiologists) we continued the studies with the biological potential of the extracts. Also, the year 2019 was normal from the climatic point of view; the following year being very rainy (was not suitable for the study).

  1. There is something incomprehensible in paragraph 2.5: Has chlorophyll actually been estimated in the pruning residue? Note that it is a one-year-old wooden cane! Where does it get chlorophyll?

Yes, chlorophylls and carotenoids were determined in 1-year-old woody canes, dried and ground. We decided to conduct the determinations regarding the assimilatory pigments when we saw that the crude extract was green. As we mentioned in the text: “Most woody plant parts possess under the peridermic or rhytidome outer layers pale-green tissues (chlorenchyma) containing assimilatory pigments.” (https://doi.org/10.1007/s00114-002-0309-z).

  1. The results in the paragraph 3.2.5 are not mentioned in the materials and methods (how they were carried out).

The information is mentioned in the subsection 2.4. Extraction procedures: “The application of ultrasounds (ultrasonic bath EMMI-04D, 42 KHz, Emag, Germany) at constant temperature (25 °C) was also tested (3, 6, 9, 12 and 15 min).” The ultrasound treatment was done on glass containers containing the extraction mixture.

  1. Separate the results section from the discussion so that the paragraphs discussed are clear.

Thank you very much for your suggestion. Separation of the Results and Discussions sections is a point that can change the entire structure of the article and we do not know if it is really helpful. The fact that these sections can be presented together (along with the larger number of pages accepted) was one of the main reasons why we chose to send our study to the prestigious journal Horticulturae.

Thank you very much for your advice and comments.

Reviewer 3 Report

General Comment:

This manuscript focused on establishing a extraction of polyphenol as well biological potential of grapevine canes which are often unused. This included the preferred methods of extraction, concentrations of flavonoids and non-flavonoids, as well as amount of antioxidant and antiradical activities comparison to crude extracts and antimicrobial effect on gram-positive bacteria. The study is comprehensive and compact, however there are minor things needed to complete before publication.

Abstract:

The abstract is clear and has presented the findings well. However, the aim needs to indicate why is unused waste from grapevine pruning needs to be utilized. Is it because of the higher demand of phenolic compounds? Economic loss? To whom this study aimed for? Greener industry?

Introduction:

·        Please rearrange the introduction into the following:

o   Paragraph 1 - Problem Statement (can combine first two paragraphs to talk about the wasted potential of grapevine canes in addition to its high production especially in Romania)

o   Paragraph 2 - Conventional Treatment (can take part of paragraph 2 about what the grapevine canes are used, touch on polyphenolic compounds present in grapevine canes)

o   Paragraph 3- Proposed Treatment (talk about the extraction methods, purifications, and data scarcity regarding biological potential of purified grapevine cane)

o   Paragraph 4- Aims and Hypothesis (Last paragraph of the current introduction can be used)

·        There should also be mention of the three different cultivars of the Vitis vinifera L. studied as shown on Figure 1.

·        Please refer to: Recent Progress and Future Perspectives for Zero Agriculture Waste Technologies: Pineapple Waste as a Case Study

Materials and Methods:

·        Please show map of the area as a figure in subsection General climate conditions of the vineyard section.

·        If possible, please provide how fine the grinded grapevine canes are.

·        The rest are well explained.

Results and Discussion:

·        This section is comprehensive.

·        Please try to compare with other type of antioxidant source of similar solvents.

·        Please refer to these:

·        Green extraction of phenolic compounds from grape pomace by deep eutectic solvent extraction: physicochemical properties, antioxidant capacity

·        Obtaining green extracts rich in phenolic compounds from underexploited food by-products using natural deep eutectic solvents. Opportunities and challenges

Conclusion:

This section is well concluded.

References:

Most references are up to date but please make sure to keep it within the 5 years of research.

Extensive editing of English language required

Author Response

Reviewer 3

Dear Reviewer,

thank you very much for your positive comments and appreciation of our work.

  1. The abstract is clear and has presented the findings well. However, the aim needs to indicate why is unused waste from grapevine pruning needs to be utilized. Is it because of the higher demand of phenolic compounds? Economic loss? To whom this study aimed for? Greener industry?

Thank you for the positive comments. Changes were made and a general objective was added, considering that the Abstract section is limited regarding the number of words.

  1. Introduction:
  • Please rearrange the introduction into the following:

o   Paragraph 1 - Problem Statement (can combine first two paragraphs to talk about the wasted potential of grapevine canes in addition to its high production especially in Romania)

o   Paragraph 2 - Conventional Treatment (can take part of paragraph 2 about what the grapevine canes are used, touch on polyphenolic compounds present in grapevine canes)

o   Paragraph 3- Proposed Treatment (talk about the extraction methods, purifications, and data scarcity regarding biological potential of purified grapevine cane)

o   Paragraph 4- Aims and Hypothesis (Last paragraph of the current introduction can be used)

The Introduction was structured according to the requirements, by connecting and/or moving the paragraphs. New text was also added.

  1. There should also be mention of the three different cultivars of the Vitis vinifera L. studied as shown on Figure 1.

Considering that the Introduction is already very long and complex, we have not added additional data about the analyzed cultivars, these being known as cosmopolitan cultivars planted on extensive surfaces all over the world. Details about cultivars are presented in the M&M section.

  1. Please refer to: Recent Progress and Future Perspectives for Zero Agriculture Waste Technologies: Pineapple Waste as a Case Study

Thank you for the suggestion, we have viewed and used the structure and ideas of the suggested article.

  1. Materials and Methods:
  • Please show map of the area as a figure in subsection General climate conditions of the vineyard section.

Considering that the spatial coordinates were given (northeast of Romania, 47°10’ northern latitude and 27°35’ eastern longitude) and the large number of images and figures, we don't know if a picture with the location helps in the context of the article. Thank you very much for the suggestion and if necessary, we will additionally add a map of the location.

  1. If possible, please provide how fine the grinded grapevine canes are.

In the subsection 2.2. Biological material is mentioned that the canes were “…cut into cylinders of 0.5-0.8 cm, ground using an electrical grinder (GT110838, Tefal, France), passed through a 0.5 mm sieve (particle size < 0.5 mm).”

  1. 7. Results and Discussion:
  • This section is comprehensive.
  • Please try to compare with other type of antioxidant source of similar solvents.
  • Please refer to these:
  • Green extraction of phenolic compounds from grape pomace by deep eutectic solvent extraction: physicochemical properties, antioxidant capacity
  • Obtaining green extracts rich in phenolic compounds from underexploited food by-products using natural deep eutectic solvents. Opportunities and challenges

Thank you for the suggested articles, unfortunately we did not use DES (deep eutectic solvents) and cannot compare the results in this sense, especially since comparative studies on the analyzed plant material and solvents were provided. Of course multiple comparisons can be added.

  1. Conclusion:

This section is well concluded.

References: Most references are up to date but please make sure to keep it within the 5 years of research.

Thank you very much for the suggestions, the effort and the time allocated to review the article.

Reviewer 4 Report

After carefully reading the manuscript entitled: " Physico-chemical Characterisation, Phenolic Compounds Extraction and Biological Activity of Grapevine (Vitis vinifera L.) canes", it can be concluded that the authors spent a lot of time and effort conducting experiments and writing an article. The paper is nicely written and of good quality. However, a few things could be improved. Below are remarks and suggestions.

1.     Editing of the English language, grammar, and spelling is required.

2.     Some sentences are too long and difficult to understand.

3.     17% of plagiarism has been detected.

4.     Some phrases and sentences repeat throughout the text.

5.     The methods for some Physico-chemical parameters are outdated

6.     Units should be harmonised through the text. For example, cm3, ml, and mL could be found in the text.

7.     For antimicrobial activity, only two strains have been used. It is too little to talk about antimicrobial activity. In addition, the methodology is outdated. It is not of any significance.

8.     The HPLC-PAD profile of the grapevine (V. vinifera L.) canes purified polyphenolic extracts (Figure 6.) is not visible and is of bad quality. HPLC diagrams of standards and samples should be added as an additional file in high resolution. Furthermore, spiking should be done to be certain of the peaks in the diagram. It is a technique used to evaluate an analytical procedure's performance when testing a specific sample type. In other words, a matrix spike test helps answer whether the results are good (valid).  

9.     Based on the above, the paper needs minor revisions to be published.

1.     Editing of the English language, grammar, and spelling is required.

2.     Some sentences are too long and difficult to understand.

Author Response

Reviewer 4

Dear Reviewer,

thank you very much for your positive comments and appreciation of our extensive work.

  1. After carefully reading the manuscript entitled: " Physico-chemical Characterisation, Phenolic Compounds Extraction and Biological Activity of Grapevine (Vitis viniferaL.) canes", it can be concluded that the authors spent a lot of time and effort conducting experiments and writing an article. The paper is nicely written and of good quality. However, a few things could be improved.

Thank you for your appreciation, remarks and useful comments on the text.

  1. Some sentences are too long and difficult to understand.

Thank you for your observations, we tried to make it clear and express the meaning of the text as best as possible.

  1. 17% of plagiarism has been detected.

All the data from the specialized literature was cited accordingly (as see in the References section).

  1. Some phrases and sentences repeat throughout the text.

We reread the manuscript and probably your observation refers to text from the Abstract, Introduction or Conclusions, where indeed some words are repeated as necessary.

  1. The methods for some physico-chemical parameters are outdated

The physico-chemical characteristics (diameter, moisture, total dry matter and minerals) were determined according to the current international standards methodology, while starch and sugars (carbohydrates) were assessed according to a Romanian official standard in force.

  1. Units should be harmonised through the text. For example, cm3, ml, and mL could be found in the text.

Thank you for this observation, we used cm3 to present the volume of the flasks (for the available space) and mL to express the sample volume used (for liquids). ml (appears only once) and has been corrected as mL.

  1. For antimicrobial activity, only two strains have been used. It is too little to talk about antimicrobial activity. In addition, the methodology is outdated. It is not of any significance.

Due to the limitations of the microbiology laboratory, two pathogenic strains were actually tested (G+ and G-). We believe that the idea was formed and the study can represent a starting point for future studies. Moreover, disk diffusion susceptibility testing is a standardized technique widely used for anti-bacterial compounds in routine clinical microbiology laboratories (https://doi.org/10.1128/AAC.01373-10; https://doi.org/10.1016/B978-0-12-803642-6.00012-5).

  1. The HPLC-PAD profile of the grapevine (V. vinifera L.) canes purified polyphenolic extracts (Figure 6.) is not visible and is of bad quality. HPLC diagrams of standards and samples should be added as an additional file in high resolution. Furthermore, spiking should be done to be certain of the peaks in the diagram. It is a technique used to evaluate an analytical procedure's performance when testing a specific sample type. In other words, a matrix spike test helps answer whether the results are good (valid).

Unfortunately, we do not have the possibility to use the proposed technique (matrix spike), the chromatograms were obtained on a standard liquid chromatograph (HPLC-DAD), using external standards. Individual compound concentrations were calculated according to phenolic compound standards and calibration curves. In the future we will try to use the spike technique for a better representation of the peaks, thank you for the suggestion.

  1. Based on the above, the paper needs minor revisions to be published.

Thank you very much for the final appreciation, it encourages us to go further, and to carry out more and more interesting and complex studies for the scientific community.

Round 2

Reviewer 2 Report

Thanks to the authors for answering all the comments.

Minor editing of English language required

Reviewer 3 Report

The manuscript is corrected and revised according to the reviewer's comments. I am now satisfied with the new version, so I would like to recommend its publication.

Minor editing of English language required